# Phylogenetic evidence reveals early Kra-Dai divergence and dispersal in the late Holocene

Yuxin Tao[1,12], Yuancheng Wei[2,12], Jiaqi Ge[3,12], Yan Pan[4,12], Wenmin Wang[5], Qianqi Bi[6], Pengfei Sheng[7], Changzhong Fu[5], Wuyun Pan[8,9], Li Jin ®[1], Hong-Xiang Zheng ®[10] ✉ & Menghan Zhang ®[8,10,11] ✉

Studying language evolution brings a crucial perspective to bear on questions of human prehistory. As the most linguistically diverse region on earth, East and Southeast Asia have witnessed extensive sociocultural and ethnic contacts among different language communities. Especially, the Kra-Dai language family exhibits tremendous socio-cultural importance in these regions. Due to limited historical accounts, however, there are several controversies on their linguistic relatedness, ambiguities regarding the divergence time, and uncertainties on the dispersal patterns. To address these issues, here we apply Bayesian phylogenetic methods to analyze the largest lexical dataset containing 646 cognate sets compiled for 100 Kra-Dai languages. Our dated phylogenetic tree showed their initial divergence occurring approximately 4000 years BP. Phylogeographic results supported the early Kra-Dai language dispersal from the Guangxi-Guangdong area of South China towards Mainland Southeast Asia. Coupled with genetic, archaeological, paleoecologic, and paleoclimatic data, we demonstrated that the Kra-Dai language diversification could have coincided with their demic diffusion and agricultural spread shaped by the global climate change in the late Holocene. The interdisciplinary alignments shed light on reconstructing the prehistory of Kra-Dai languages and provide an indispensable piece of the puzzle for further studying prehistoric human activities in East and Southeast Asia.

East and Southeast Asia host several great ancient civilizations and the world's most populous countries, exhibiting great genetic and socio-cultural diversities[1–4]. Language is the carrier of socio-cultural activities and is often shaped by complex demographic dynamics[5]. Therefore, studying language evolution brings a crucial perspective to bear on

questions of human prehistory. However, ethnolinguistic prehistory remains poorly understood in East and Southeast Asia.

In these regions, the Kra-Dai language family (also known as Tai-Kadai) is spoken by nearly 100 million people and geographically distributed in a vast region encompassing South China, Mainland

[1]State Key Laboratory of Genetic Engineering, Center for Evolutionary Biology, Human Phenome Institute, Zhangjiang Fudan International Innovation Center, School of Life Science, Fudan University, Shanghai 200438, China. [2]School of Chinese Language and Literature, Guangxi Minzu University, Guangxi Zhuang Autonomous Region, Nanning, China. [3]Department of Chinese Language and Literature, Fudan University, Shanghai, China. [4]Department of Cultural Heritage and Museology, Fudan University, Shanghai, China. [5]College of Nationalities, Guangdong Polytechnic Normal University, Guangzhou, China. [6]College of Communication, East China University of Political Science and Law, Shanghai, China. [7]Institute of Archaeological Science, Fudan University, Shanghai, China. [8]Institute of Modern Languages and Linguistics, Fudan University, Shanghai, China. [9]Institute for Humanities and Social Science Data, School of Data Science, Fudan University, Shanghai, China. [10]Ministry of Education Key Laboratory of Contemporary Anthropology, Department of Anthropology and Human Genetics, School of Life Sciences, Fudan University, Shanghai, China. [11]Research Institute of Intelligent Complex Systems, Fudan University, Shanghai, China. [12]These authors contributed equally: Yuxin Tao, Yuancheng Wei, Jiaqi Ge, Yan Pan. ✉e-mail: zhenghongxiang@fudan.edu.cn; mhzhang@fudan.edu.cn

Southeast Asia (MSEA), and Northeast India[6,7]. Their geographic distributions are surrounded by or intermingled with the settlements of the four other language families: Austronesian, Austroasiatic, Sino-Tibetan, and Hmong-Mien (Fig. 1a)[6]. The accumulated linguistic surveys reveal the predominance of Kra-Dai languages in the contact-induced convergence of linguistic structures of other languages in the MSEA *sprachbund*[8]. Kra-Dai languages thus show tremendous socio-linguistic importance in this *sprachbund* and gradually become major areal vectors for cultural, economic, and political life in the past 2000 years[8]. Therefore, understanding the prehistory of Kra-Dai languages plays a crucial role in uncovering their complex demographic dynamics and socio-cultural interactions with surrounding ethnic populations in South China and MSEA.

Three fundamental issues remain dubious and then hamper reconstructing the history of the Kra-Dai languages. The first issue is linguistic relatedness. The Kra-Dai languages primarily comprise five well-described branches: Kra, Hlai, Ong-be, Tai, and Kam-Sui. However, their relationships are still controversial. The foremost one is the position of Kra. Some scholars advocate the Kra languages as a primary branch in the Kra-Dai language family while others suppose that Kra languages should be below the Kam-Sui group[8,9]. Similarly, the placement of Hlai is also an ongoing debate whether Hlai is a primary branch or a lower position as a sister of Ong-Be and Tai[9,10]. Moreover, it remains no consensus on the explicit relations of Ong-Be with the Tai and Kam-Sui branches, respectively[9,11]. In addition, there are ongoing debates on the low-level branches of Tai languages such as the division between the Central and Southwestern branches on a par with the Northern branch[12-15]. Different proposals on Kra-Dai classification have different implications for the topological

structure of Kra-Dai phylogeny ranging from more rake-like to more hierarchically nested[16].

The second issue is the time-depth of the Kra-Dai language divergence. The demographic documents can only provide clues about the Kra-Dai language dispersal from the Southern China region to Thailand and Laos in the past 750 years[8]. However, the initial divergence time of the Kra-Dai languages is ambiguous due to lacking available prehistoric records. In historical linguistics, some scholars regard the Kra-Dai language family as a very old phylum and speculate their initial divergence to have occurred 5000–6000 years ago[17,18]. In contrast, other scholars consider that the first split of Kra-Dai languages could take place no more than 4000 years before the present (BP)[19,20]. Despite ambiguities on the Kra-Dai language divergence time, the recent phylogenetic studies of the surrounding language families such as Austronesian[21] and Sino-Tibetan[22,23] illustrate the vitality of language evolution in MSEA and adjacent Southern China region 5000–6000 years ago. Accordingly, most Chinese linguists favor that in the same geographic area, the initial divergence of Kra-Dai languages could be traced back to at least 5000 years BP[17,18].

The third issue is the dispersal routes of Kra-Dai languages. According to the traditional view of *Urheimat* inference in linguistics, the Kra-Dai dispersal can originate from the Guizhou inland of China which exhibits the highest linguistic diversity among the present Kra-Dai languages[18]. And then, the Kra-Dai languages dispersed southward into Guangxi, Hainan Island, and MSEA; and simultaneously westward into Yunnan inland and further into MSEA. This scenario can be summarized as the Inland Origin Hypothesis of Kra-Dai languages (Fig. S1a). Alternatively, a more widely accepted view suggests that the homeland of Kra-Dai languages could be laid in coastal south China, possibly the

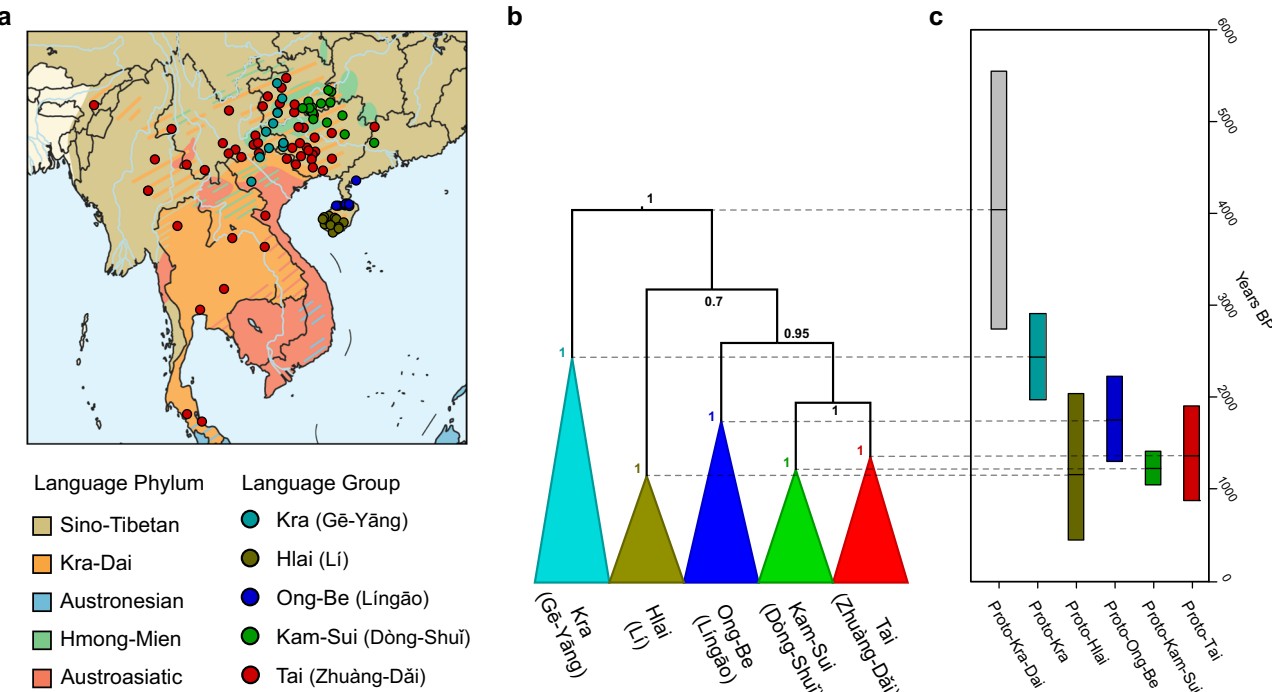

**Fig. 1 | The geographical distribution and the maximum clade credibility tree with a divergence time of the Kra-Dai languages. a** The geographical distribution of 100 Kra-Dai language samples and the language phyla in South China and Mainland Southeast Asia. The base map was derived from an R package *rnaturalearth* (URL: https://github.com/ropensci/rnaturalearth). **b** The maximum clade credibility tree was shown with posterior values. The clades were collapsed by language groups. Each language branch was assigned with a specific color. **c** The bar plot was shown for the 95% highest posterior density (HPD) for the divergence time estimations of six Proto languages in Kra-Dai languages. Data were derived

from an MCMC run of 9000 posterior samples (more details see Fig. S5b and Supplementary Data 9). Proto-Kra-Dai (unit: years BP): min = 2158, max = 9023, mean = 4041, 95% HPD = 2741–5550; Proto-Kra: min = 1625, max = 3317, mean = 2435, 95% HPD = 1967–2909; Proto-Hlai: min = 291, max = 3000, mean = 1155, 95% HPD = 443–2035; Proto-Ong-Be: min = 803, max = 2713, mean = 1750, 95% HPD = 1299–2226; Proto-Kam-Sui: min = 896, max = 1579, mean = 1222, 95% HPD = 1044–1410; Proto-Tai: min = 592, max = 2200, mean = 1360, 95% HPD: 873–1903.

area of Fujian, Guangdong, and Guangxi provinces[24], and their dispersal routes exhibited a radial expansion into Guizhou-Yunnan inland, Hainan Island, and MSEA, respectively. It can hence be regarded as the Coastal Origin Hypothesis of Kra-Dai languages (Fig. S1b). Apart from linguistics, both archaeological and genetic studies may provide a broader hypothetic scenario of the demic and cultural diffusion in South China and MSEA. In particular, the archaeological assemblages such as the development of food production depict an overall picture of the spread of Neolithic human populations from the Central and Southern China region throughout the vast regions of MSEA[25]. The genetic analyses of ancient DNA also identify the genetic ancestries of MSEA located in South China. The indigenous people in MSEA admixed with multiple waves of migration from South China in the last 4000 years[2,26]. These genetic findings are compatible with the archaeological observations on the later migrations from South China into the MSEA, especially 2000 years ago. They also indicated that the MSEA should have experienced unprecedented demographic and cultural changes in prehistory[8,27].

Before addressing the latter two issues, the key foundation is to reconstruct the explicit genetic relationships among Kra-Dai languages at first. In historical linguistics, lexicostatistics is a quantitative method to estimate the percentage of lexical cognates between languages and then determine their genetic relatedness. As an important application of lexicostatistics, glottochronology is further proposed to estimate approximate separation dates between two languages based on the proportion and rates for morpheme replacement in a relatively stable basic vocabulary[28,29]. However, glottochronology has been roundly criticized due to its improper methodological assumptions such as the constant rates of language change or morpheme replacement. And this approach ignores the methodological sensitivity to contact-induced lexical borrowings[30]. Moreover, lacking sufficient historical accounts and comprehensive investigations of Kra-Dai languages is also a stumbling block to the applications of lexicostatistics and glottochronology. Due to the conceptual similarities between language and biological evolutions[31], recent advances in Bayesian phylogenetic methods derived from evolutionary biology shed light on the reconstruction of the language family tree under considerations of rate variations of languages and finite categories of words[21–23,32–34]. Moreover, aligning the interdisciplinary evidence from linguistics, genetics, and archaeology is another powerful tool to provide a comprehensive landscape for the demic and cultural diffusions of the Kra-Dai-speaking populations[8].

In this study, we utilize Bayesian phylogenetic methods to reconstruct the prehistory of Kra-Dai languages. For this purpose, a large lexical database has been employed which consists of 90 basic lexical items from 100 Kra-Dai languages. The basic lexical items are derived from the Swadesh 100-word list. And then we use advanced computational methods to reconstruct the linguistic relatedness, estimate the divergence time, and infer the dispersal routes. Furthermore, we find that the evolution of the Kra-Dai language was related to the dynamic changes in the natural environment and socio-cultural scenarios in which the Kra-Dai-speaking populations resided. Accordingly, we align diverse interdisciplinary data from genetics, archaeology, paleoecology, and paleoclimatology. This interdisciplinary alignment can provide valuable insights into the prehistory of Kra-Dai languages and has enabled us to better comprehend the demic and cultural history in East and Southeast Asia.

## Results
### Compiling the lexical cognate database of Kra-Dai languages
To establish the lexical database of Kra-Dai languages, we gathered and integrated lexical data from previously published literature and our first-hand linguistic fieldwork. The collected language samples are geographically distributed across South China, MSEA, and Northeast India (Fig. 1a). The sample size in this dataset is larger than that of the

languages named as Kra-Dai or Tai-Kadai in Glottolog[35] and Ethnologue[36] databases. Based on the Swadesh 100-word list[37], we manually assembled a vocabulary list from different bibliographies and identified their cognateness based on regular sound correspondences under the framework of historical comparative method in linguistics (Supplementary Data 1, Supplementary Information section 1.1, and section 1.2). These cognate sets were then numerically coded into the binary state for each where 1 represented the presence of a specific cognate, 0 represented its absence, and '?' was provisional. Finally, we identified 646 lexical cognate sets of 90 basic lexical items for 100 Kra-Dai languages (Supplementary Data 1).

### Reconstructing the dated Bayesian phylogenetic tree of Kra-Dai languages
Using our lexical database, we conducted a Bayesian phylogenetic analysis to reconstruct the relatedness of Kra-Dai languages. To estimate the time depth of language divergence, we specified several time calibrations based on available linguistic documents and ethnic archives (Supplementary Data 2). We compared six model combinations with different parametric settings using logarithmic Bayes Factor (log BF), and found that the combination of the covarion model and relaxed lognormal clock model was the best-fitting (Fig. S2 and Supplementary Data 3). To avoid any artificial bias, we performed the phylogenetic reconstruction without any ancestral or monophyletic constraints as priors.

The Bayesian phylogenetic reconstruction showed that the classification of Kra-Dai languages consisted of five well-established branches (Figs. 1b and S3, S4). All these branches were monophyletic and supported by high posterior probability values for each (Figs. 1b and S3, S4). Specifically, the Kra and Hlai languages branched off successively from other languages of the Kra-Dai language family. Ong-Be was a sister of the cluster of Tai and Kam-Sui branches. The Tai branch was further divided into three groups: Northern Tai, Central Tai, and Southwestern Tai. The inferred language relationships among these five branches were consistent with Ostapirat's classification[11]. The estimated divergence time indicated that the first split of Kra-Dai languages occurred around 4000 years ago (mean value = 4041 years BP), with a 95% HPD interval range of approximately 2700 to 5500 years ago (Figs. 1c and S5). The estimated time was significantly lower than Liang, Zhang, and Li's expectation of Kra-Dai divergence over 5000 years ago[17,18] (t = −119.41, p-value < 2.2e−16, Fig. S5a), but was largely consistent with Ostapirat and Peiros's inference[19,20]. The initial divergence time estimations of Kra-Dai languages under different model combinations were compatible with each other (Fig. S6).

To examine the two dispersal hypotheses of Kra-Dai languages, we then conducted discrete phylogeographic inference using the Bayesian phylogenetic comparative approaches. The geographic distribution of Kra-Dai language samples was categorized into five distinct areas: the Guangxi-Guangdong coastal area, two separated inland areas of Yunnan and Guizhou provinces, the island area of Hainan province, and the MSEA covering other areas including Thailand, Vietnam, Laos, Myanmar, and India in this study, respectively. Given the reconstructed Kra-Dai language phylogeny, we performed the ancestral state reconstruction and found that the coastal area was the most probable origin of Kra-Dai languages, with a maximum probability of 47.0%, which was significantly higher than those probabilities of other areas (Figs. 2 and S7 and Supplementary Data 4). This result supported the Coastal Origin Hypothesis proposed by Gong[24]. We further evaluated five model combinations with different transition states among the five areas to infer the dispersal routes of Kra-Dai languages using the Bayesian reversible-jump Markov Chain Monte Carlo method (RJ-MCMC)[38]. The best model with the highest Bayes Factor (BF = 42.67) strongly supported the scenario that no transitions occurred between non-adjacent areas geographically and none between MSEA and Hainan Island (Table 1 and Fig. S8). Therefore, our

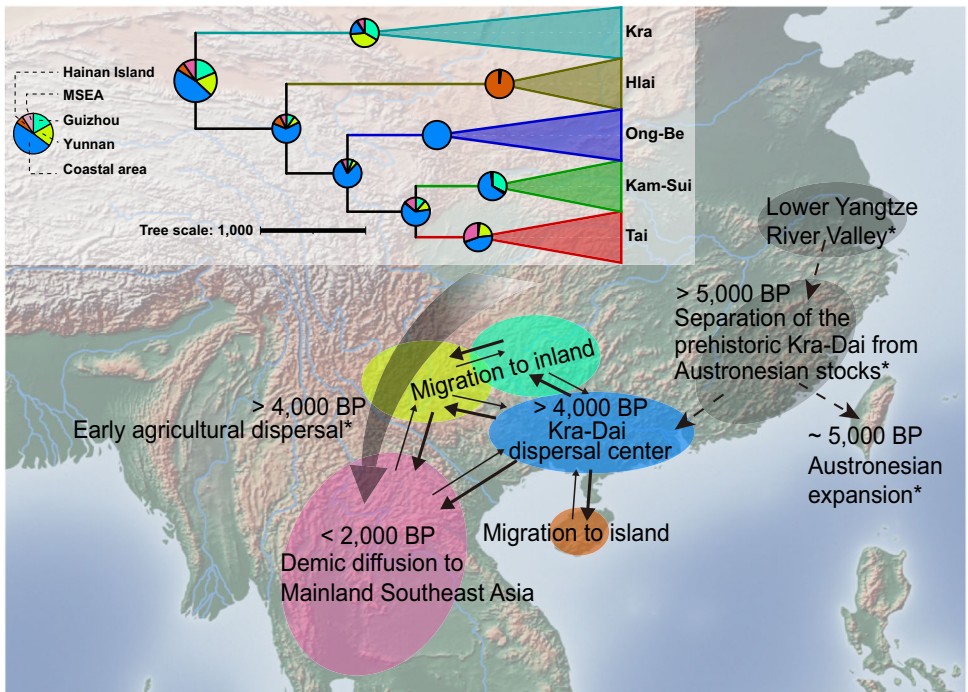

**Fig. 2 | The inferred dispersal routes of Kra-Dai speakers and their languages in prehistory.** The phylogenetic tree was drawn based on the MCC tree of Kra-Dai languages. Each pie chart on internal nodes of the Kra-Dai tree showed the posterior possibilities of each geographic area estimated by the Reversible jump Markov chain Monte Carlo method implemented in *BayesTraits*. The isolated pie chart showed the posterior possibilities for each geographic candidate on the root of the Kra-Dai tree (possibilities for Guizhou: 18.6%, Yunnan: 18.3%, Coastal Area: 47.0%, Hainan Island: 6.5%, and the MSEA: 9.6%). On the map, the dashed lines and symbols of star shapes represented the conclusions derived from previous genetic studies[1,2,26,44–46,53,54]. The solid lines represented our inference, and their thickness was in proportion to the values of transition rates between each pair of geographic areas estimated by discrete phylogeographic analysis (Fig. S9). The great cycle shapes with different colors represented the geographic areas: red for the Coastal area (Guangxi-Guangdong), green for Guizhou inland, yellow for Yunnan inland, brown for Hainan Island, and pink for MSEA. The estimated time was denoted on the map. The base map was derived from the vector map data from https://www.naturalearthdata.com.

results illustrated that some early Kra-Dai languages spread across the Qiongzhou Strait and into Hainan Island; some expanded northwestward into the inland areas of Yunnan and Guizhou provinces, and further southwestward into MSEA; and some dispersed into MSEA directly from the coastal area (Figs. 2 and S9).

## Aligning the interdisciplinary evidence for Kra-Dai language prehistory

To provide a more comprehensive understanding of the social and cultural context surrounding the Kra-Dai language divergence and dispersal, we integrated interdisciplinary evidence from genetics,

**Table 1 | Comparison of models of dispersal routes tested in this study**

| Name | Descriptions | BF |
|---|---|---|
| FULL | Allowing transitions between any areas | 0 |
| Model 1 | Only transitions between geographically adjacent areas | 6.69 |
| **Model 2** | **Model 1 + No transition between MSEA and Hainan Island** | **42.67** |
| Model 3 | Model 1 + No transition between MSEA and coastal area | 17.49 |
| Model 4 | Model 1 + No transition between MSEA and Yunnan inland | 0.05 |

Bayes factors (BFs) are calculated based on the number of times the RJMCMC visits a particular model of evolution in comparison to the expected number of times in all 3 runs (Supplementary Information section 1.5, Fig. S8 and Supplementary Data 10). BF < 1 is evidence against a model. BF > 30 is very strong evidence in favor of a model. The bold highlights Model 2 with the maximum BF value.

archaeology, paleoecology, and paleoclimatology to depict the evolutionary process of Kra-Dai languages. As illustrated in Fig. 3, the divergence tempo of Kra-Dai languages showed that the initial divergence occurred at ~4000 years BP and the second one occurred at ~3200 years BP, then the language numbers increased continuously in the past 2300 years (Figs. 3a and S10). According to archaeological evidence, the number of archaeological sites in Southern China decreased dramatically at ~4000 years BP, then increased and reached its maximum at ~3000 years BP (Fig. 3b). The genetic evidence was represented by the Bayesian Skyline Plot of the Kra-Dai mtDNA lineages which reflected the historical change of Kra-Dai population size. Generally, we found two phases of population growth, of which the former was an approximately 17-fold demographic increase during 6400–4200 years BP and the latter was an approximately 16-fold demographic leap from 3500 years BP till now (Fig. 3c). In addition, the paleo-ecological evidence suggested that the survival probabilities of tropical rice decreased dramatically in eastern China and high-altitude southwestern China during 4400–3500 years BP and then maintained a relatively stable[39] (Fig. 3d). Lastly, based on the paleo-climatological evidence[40–42], we found the global temperature decrease known as the 4.2 K event, which took place from 4400 to 3500 years BP and minimum at ~4000 years BP. Then, the global temperature became relatively stable in the past 3000 years (Fig. 3e).

Accordingly, we could summarize the evolutionary history of Kra-Dai languages into three periods. The first one was the "contraction period" during the 4.2 K event (4400–3500 years BP), coupled with the initial divergence of Kra-Dai languages, a nearly unchanged population size, decreasing archaeological sites, survival probabilities of rice, and temperature. The second one was the "recovery period" after the 4.2 K event (3500–2300 years BP), corresponding to the early divergence

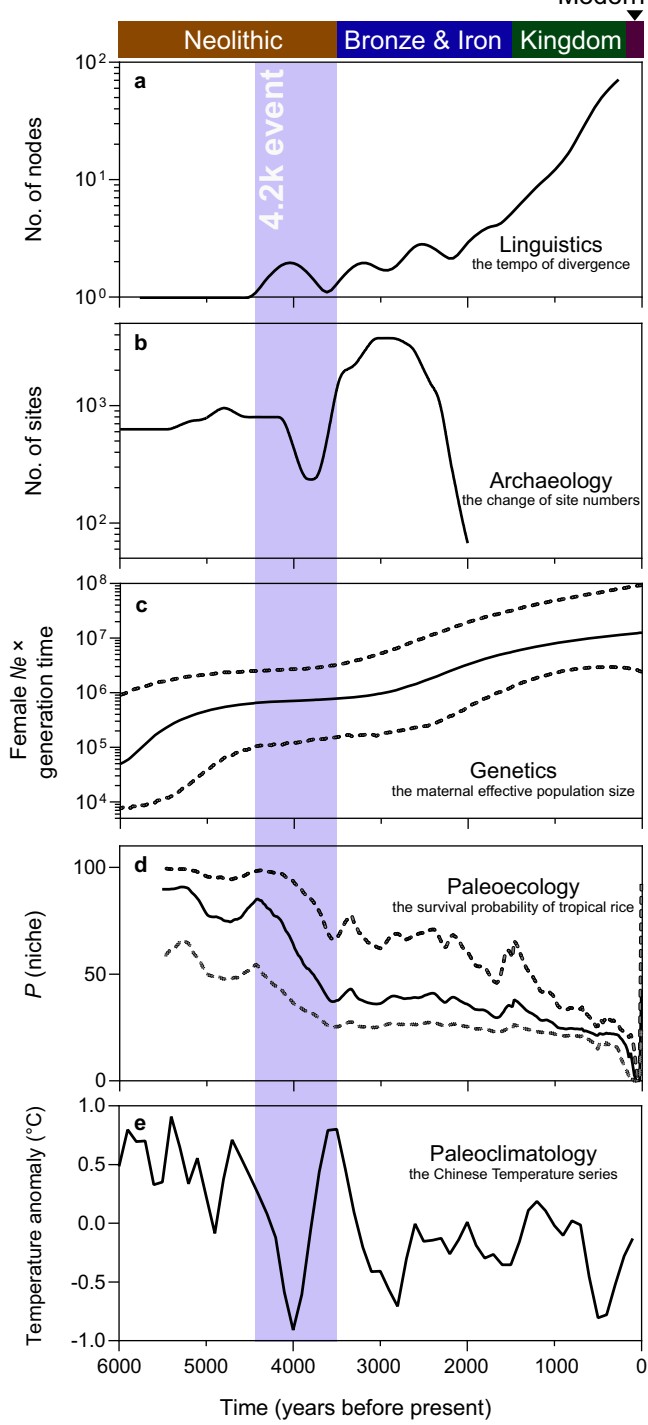

Fig. 3 | Temporal alignment of Kra-Dai language dispersal, population expansion of maternal lineages, and the dynamic changes of the paleoenvironmental contexts in the past 6000 years. a Linguistics: the tempo of the divergence of Kra-Dai languages (based on 100 languages) in the Kra-Dai Bayesian phylogeny. b Archaeology: Changes in the number of archaeological sites mapped in the region of Zhejiang, Fujian, Guangdong, Hunan, and Yunnan (n = 4816 sites). These archaeological sites dated between 9000 and 2000 years BP. Data were taken from Hosner et al. [78]. c Genetics: the Bayesian skyline plot for the maternal population was established by 22 representative mtDNA lineages of Kra-Dai-speaking samples. Solid line was the median value. Dash lines were the upper and lower bounds. d Paleoecology: the percent probability of tropical rice being in the thermal niche (assuming a requirement of 2900 growing degree days at 10 °C bases) over time. Data were taken from Gutaker et al. [39]. Solid line was the mean value. Dash lines were the 1σ uncertainty interval. e Paleoclimatology: Chinese Holocene Temperature Series. Data were taken from Fang and Hou[41]. The purple shadow highlighted the 4.2 K event (4400–3500 years BP). The division of the historical timeline in Southeast Asia was based on http://afe.easia.columbia.edu/.

phylogeographic inference suggested that the coastal area (Guangxi-Guangdong provinces) was likely the dispersal center of Kra-Dai languages, and profiled the north-south and east-west dispersal routes which were consistent with previous genetic and cultural evidence[3,4,24,26,43–46]. Furthermore, our interdisciplinary alignment revealed that the Kra-Dai language dispersal might be associated with environmental change and demographic activities in East and Southeast Asia. Overall, our findings offered a new perspective on the evolutionary dynamics of Kra-Dai languages and their contributions to shaping language diversity in East and Southeast Asia. By studying the evolutionary history of Kra-Dai languages, we could gain more insights into the present socio-cultural landscape and better understand the prehistory of these regions.

Although the linguistic relationships among the five Kra-Dai branches were consistent with traditional linguists' views, some placements of specific language samples could be observed in the low-level branches which were not entirely in line with expectations (Fig. S4 and Supplementary Information section 2.2). For example, the Saek language, which was a Northern Tai language, was grouped as the sister of the Southwestern Tai group. Accordingly, four possibilities might lead to such misplacement. The first one was that the maximum clade credibility (MCC) tree in Fig. S4 might not present an accurate topological structure for all nodes. Second, the classification was given only based on lexical cognate data but not phonological or morphological traits. Third, the Swadesh 100-word list could not provide sufficient resolutions to distinguish closely related languages. Fourth, the Saek language could have experienced substantial horizontal borrowings from its surrounding Southwestern Tai languages[47]. These possibilities have been regarded as deep problems with linguistic analysis that undermine Bayesian phylogenetic methods[48] (Supplementary Information section 2.5). To address these problems, we performed a four-point analysis to examine which possibilities led to the misplacement[22] (Supplementary Information section 2.1). The result of our analysis favored that Saek should be a Northern Tai language, with about 37% of its lexical cognates influenced by Southwestern Tai languages horizontally. This proportion seemed to exceed the 20% limit that Bayesian phylogenetic methods allowed[31]. Accordingly, we suggested that this misplacement could be attributed to the methodological inadaptability for classifying borrowing-prone languages. In other words, Bayesian phylogenetic methods could generate an erroneous classification for the specific language clades and be questioned the robustness of the overall shape of the Kra-Dai phylogeny.

To examine the robustness of linguistic relatedness of the five language branches, we accordingly replicated our computational procedures with different settings of monophyletic constraints on low-level branches which conformed to different traditional linguistic

events of Kra-Dai languages, a more temperate climate than before, and a steady increase of archaeological sites and population size. The third one was the "prosperity period" (2300 years BP−the present), which witnessed a rapid increase in language numbers and population size (Supplementary Information section 2.8).

## Discussion

Studying the spatiotemporal evolution of Kra-Dai languages is crucial for comprehending the demographic activities and socio-cultural development in East and Southeast Asia. In this study, we employed Bayesian phylogenetic methods to reconstruct the linguistic relatedness of the five branches in accordance with that proposed by Ostapirat[11]. We also estimated the initial divergence of Kra-Dai languages occurring approximately 4000 years BP. The Bayesian

views. As a result, we showed that Bayesian phylogenetic methods could yield robust results supporting our conclusions regarding linguistic relatedness (Fig. S11 and Supplementary Information section 2.5). Meanwhile, the time-depth and phylogeographic inference were also robust under different low-level branching patterns (Fig. S12, Supplementary Data 5, and Supplementary Information section 2.5). Moreover, the results of reticulate signal detection suggested that the Kra-Dai languages exhibited patterns of linguistic isolation at the early divergence stage which could be induced by population migrations, and then the language contacts could be found in the low-level branches (Supplementary Data 6 and Supplementary Information section 2.12). These findings supported our postulated evolutionary scenarios for Kra-Dai languages and were in favor of the traditional view of the human population as the carrier of languages[49].

Furthermore, we observed the strong coupling of the linguistic and demographic dynamics with the changes in the paleoenvironmental context (Figs. 3 and S10). In general, the paleoenvironmental context consists of the paleoecologic and paleoclimatic factors which are regarded as crucial drivers to shape the demographic activities of prehistoric populations[50–52]. Synthesizing the interdisciplinary evidence, we proposed a possible scenario that prehistoric Kra-Dai language divergence and dispersal accompanying population expansion could be driven by the dynamic changes in the paleoenvironmental context (Supplementary Information section 2.7 and section 2.8). In particular, as early as 5000 years BP, the rice farmers in the lower Yangtze River Valley, namely, Bai Yue nationalities, were divided into Kra-Dai-speaking and Austronesian-speaking populations, respectively[1,2,26,44–46,53,54]. During the "contraction period", the Kra-Dai-speaking populations were forced to experience the migration process and population divergence in the deteriorating environment. This process induced the initial Kra-Dai language divergence. Due to the collapse of agriculture and the shortage of food, some settlements were abandoned, resulting in a decrease in archaeological sites; meanwhile, the number of Kra-Dai-speaking populations of maternal lineages grew slowly, indicating that the population size might maintain nearly unchanged. In the "recovery period", the temperature did not fluctuate dramatically, and food production became more stable than before. This situation promoted the steady growth of the population size of Kra-Dai-speaking populations, and people started to migrate actively and more frequently to find more settlements. Such population activities in the "recovery period" also resulted in the early language divergence events. These findings suggested that the prehistoric divergence of Kra-Dai languages might be coupled with the climate-induced demographic activities (e.g., migration) of Kra-Dai-speaking populations. In contrast, during the "prosperity period", the long-term stable temperature and food production allowed the number of Kra-Dai languages and the size of Kra-Dai-speaking populations to increase spontaneously, contributing to more frequent demographic activities such as population expansions and interactions. (Figs. 3 and S10, Supplementary Information section 2.7, and section 2.8).

Specifically, the spatiotemporal coupling of the prehistoric Kra-Dai language dispersal and agricultural spread was in favor of the language/farming dispersal hypothesis[55]. This hypothesis proposed a connection between the spread of languages and farming in prehistoric periods[55,56]. According to the hypothesis, the spread zone of Kra-Dai languages could be attributed to the spread of rice-dominant mixed farming in South China[56]. In most cases, the driving force was agricultural prosperity. However, a different scenario of environmental changes was observed during the 4.2 K event (i.e., the collapse of agriculture and harsh environment). Based on our interdisciplinary analysis, the Kra-Dai language and agricultural spread might be driven by the climate fluctuating and the survival probability of rice declining dramatically[39,41,42] (Fig. 3 and Supplementary Information section 2.9). Additionally, we deduced that the Kra-Dai language dispersal could not

be related to the early agricultural spread in MSEA. The major reason was that the agricultural records in MSEA were not completely contemporaneous with the prehistorical dispersal of Kra-Dai peoples but with the Austroasiatic ancestors[53,57–61] (Fig. S13 and Supplementary Data 7). In other words, the early agricultural spread and development in MSEA might be driven by ancient Austroasiatic people which was advocated by genetic evidence[1] (Supplementary Information section 2.9). All in all, we proposed that agricultural recession induced by climate fluctuation could be a driving force for the prehistorical co-dispersal of Kra-Dai languages and agriculture in South China (Supplementary Information section 2.9).

The evidence for inferring human history is presumed to be the considerable parallelism of archaeological remains, genetic components, and languages[5,62,63]. Accordingly, an interdisciplinary alignment is a promising approach to understanding the population prehistory involving demographic dynamics, language evolution, and cultural innovation[64–66]. As multi-ethnic areas, South China and MSEA experience substantial population activities and socio-cultural interactions from the past to the present. The complex process of population activities profiled a scenario of the demic diffusion of Kra-Dai-speaking ancestors in the vast Southern China region and MSEA (Fig. 2). The reconstructed ethnolinguistic history of Kra-Dai-speaking populations would be a foothold for studying the intricate linguistic relationship and human history of South China and MSEA. Even so, several unresolved issues are worth further and more comprehensive investigations in the future such as the relationship between Kra-Dai and Austronesian languages, and that between the Kra-Dai-speaking populations and Bai Yue nationalities[67,68].

## Methods
### Ethics statement
The collection of modern samples and the sequencing protocol have been reviewed and approved by the Human Ethics Committee of the School of Life Sciences at Fudan University (permission no. 218, 29th Feb 2012), following the ethical research principles of the Ministry of Science and Technology of the People's Republic of China (Interim Measures for the Administration of Human Genetic Resources, 10 June 1998). Study staff informed potential participants about the goals of the project, and the individuals who chose to participate gave informed consent consistent with broad studies of population history and human variation and public posting of anonymized data. There were no rewards for participating and no negative consequences for not participating; all participants signed or affixed a thumbprint to the consent form reviewed by Fudan University.

### The lexical cognate database for Kra-Dai languages
We compiled a large-scale lexical cognate database for Kra-Dai languages by gathering 100 lexical meanings from the Swadesh 100-word list and 100 language samples from Kra, Hlai, Ong-Be, Kam-Sui, and Tai branches, respectively. The database synthesized Starostin's cognate database (https://starling.rinet.ru/cgi-bin/main.cgi), several previous research reports, and first-hand language documents from our linguistic fieldwork (Supplementary Data 8). We used the traditional historical-comparative method to manually identify the lexical cognate sets, which were later cross-checked by other linguistic scholars (Supplementary Data 1, Supplementary Information section 1.1, and section 1.2). We identified 90 lexical items out of the Swadesh 100-word list that showed lexical data coverage of over 70% and represented genuine cognate sets inherited from a common ancestor of Kra-Dai languages without lexical borrowings. We coded these sets of cognates using 0, 1, and '?' to indicate their absence, presence, and uncertainty in each language sample, respectively. This resulted in 646 binary-coded cognate sets, and we added an ascertainment bias column for each first column of every lexical item to obtain the alignment[69,70]. Finally, we used 736 binary-coded data to

reconstruct the phylogeny of Kra-Dai languages (Supplementary Data 9).

## Phylogenetic reconstruction and divergence time estimation

We used the BEAST v2.6.3 program with the Babel package v0.3.1 (https://github.com/rbouckaert/Babel) to reconstruct the phylogenetic tree of Kra-Dai languages and estimate the divergence time[34,71] (Supplementary Data 9). We tested six combinations of two site models such as continuous-time Markov chains (CTMC) and covarion models, clock models including the strict and relaxed lognormal clock, and the gamma rate heterogeneity with one or four rate categories for the CTMC model. Since we have not sampled all languages and some languages may have gone extinct, and no old languages were included, we thus adopted the Birth Death Skyline Contemporary (BDSParam) model as the tree prior[70,72]. This model uses the parameters of birth rate $\lambda$ and death rate $\mu$, and creates trees starting with their root. In this model, the exponential distribution with the mean value of 0.01 was selected as the prior candidate for $\lambda$ and $\mu$, and their initial value was set to 0.01 and 0.008, respectively; and a beta distribution with $\alpha = 100$, $\beta = 19$ was selected as the prior candidate for sampling proportion $\rho$. Because we aimed to obtain an inferred phylogeny of the Kra-Dai languages, we did not set any monophyletic constraints as priors, even if they were well-attested branches. To estimate the divergence time of Kra-Dai languages, we used demographic evidence and historical records as calibrations to scale the trees (Supplementary Data 2). The six models were run for 50,000,000 generations, samples in every 5000 generations, with a burn-in of the first 1000 samples. Finally, we obtained a posterior sample size of 9000. Tracer v1.6 was used to check autocorrelation and convergence status and to test the best-fitting model combination by their likelihood value and ln Bayes factors using Harmonic Mean Estimator (Fig. S2, Supplementary Data 3). The comparison of model combinations was also performed by the Path Sampling method following the guideline (URL: http://www.beast2.org/2014/07/14/path-sampling-with-a-gui) (Supplementary Data 3). The maximum clade credibility (MCC) tree was generated by using TREEANNOTATOR v2.4.6 with a posterior probability limit of 0.5 after discarding the first 10% of the trees. DENSITREE v2.2.7[73] was applied to illustrate the variation in the posterior sample of trees (Fig. S4).

## Discrete phylogeographic inference

To infer the dispersal routes of Kra-Dai languages, we used phylogenetic comparative approaches to examine the transitions among different areas and reconstruct the ancestral area of Kra-Dai languages. Here, we divided the geographical distributions of Kra-Dai language samples into five distinct areas. These areas comprised Guizhou inland, Yunnan inland, coastal area (Guangxi-Guangdong), Hainan Island, and MSEA (other areas including Thailand, Vietnam, Laos, Myanmar, and India in this study). Accordingly, each language sampled in our study was assigned to a definite geographical area. According to the available historical ethnic records[74], we manually set the location of the most recent common ancestor of Ong-Be languages to the Guangxi-Guangdong coastal area. We performed a *Discrete* program for the multi-state model implemented in *BayesTraits* (http://www.evolution.reading.ac.uk/BayesTraitsV3.0.5/BayesTraitsV3.0.5.html) on 1000 trees. The 1000 trees were randomly resampled in all posterior sample trees generated by BEAST after a burn-in of the first 10% of samples. Given the trees, we reconstructed the ancestral area for the dispersal center of proto languages geographically. To explore the dispersal routes among different areas, we tested five models and used the Bayes Factor to choose the optimum one (Supplementary Data 10 and Supplementary Information section 1.5). To estimate the possibilities for each geographic distribution in each internal node of the Kra-Dai language phylogeny, we used *AddNode* command in the *BayesTraits* program to reconstruct a specific node on a tree if present.

These specific internal nodes were defined according to the topology of the MCC tree. We then employed a reversible-jump Markov-Chain Monte Carlo approach (RJ MCMC)[38], where the approach was run for 55,000,000 iterations, sampled in every 5000 iterations with a burn-in of the first 1000 samples. Finally, we obtained 10,000 posterior samples. Specifically, to find out the most probable homeland for Kra-Dai languages, we used paired one-side Wilcoxon signed rank test to find whether there are significant differences among the probabilities of the five distinct areas in the posterior samples (Fig. S7). For parameter settings, a hyperprior was used to seed two parameters of an exponential distribution from uniform distributions on the interval 0 to 10. The branch lengths were rescaled by a factor of 0.0001. The RJ MCMC was run 3 times to ensure that the results were stable (Supplementary Data 11 and Supplementary Data 10).

## Inferring Kra-Dai-speaking maternal population dynamics

To study the population expansion of Kra-Dai-speaking populations, we reconstructed the maternal demographic history using the mitochondrial DNA (mtDNA) data, representing maternal inheritance of the Kra-Dai-speaking populations. Six Kra-Dai-speaking populations were included and 22 representative mtDNA lineages were identified in this study. Related samples in China consisted of 27 Dong, 19 Zhuang, and 35 Dai individuals, whereas related samples in MSEA consisted of 12, 56, and 266 Kra-Dai-speaking individuals in Laos, Vietnam, and Thailand, respectively (See details and accession codes of genetic data in Supplementary Data 12).

In particular, we collected Dong subjects mainly from Hunan Province and Zhuang subjects from Guangxi Zhuang Autonomous Region in China during the investigation of Chinese ethnic population groups led by MOE key laboratory of Contemporary Anthropology of Fudan University. This research was approved by the Human Ethics Committee of the School of Life Sciences at Fudan University (permission no. 218, 29th Feb 2012), and was carried out following the approved guidelines. We followed the recommendations provided by the revised Helsinki Declaration of 2000. The participants responded to community advertising for our investigation on local ethnic groups and were recruited at the local study sites. All the samples included in this study were maternally unrelated and all of their parents were confirmed Dong or Zhuang people, respectively. The sample donors aged from 18–60 years old, and all of them were informed of an overview of the investigation and signed written informed consent before participating in the study. The details of mtDNA library preparation, sequencing, assembly, variant calling, sample selection, and sequence processing were demonstrated in Supplementary Information section 1.6.

Finally, a total of 22 mtDNA lineages (including 415 Kra-Dai-speaking samples) were identified as the representatives of Kra-Dai language expansion (Supplementary Data 12). We reconstructed the variation of the historical effective population size of the 22 mtDNA lineages via coalescent Bayesian skyline plots (BSP) implemented in BEAST v1.8 and Tracer 1.5.1[75] (Fig. S14). We used the coding regions of 269 mtDNA sequences with different haplotypes of the above 415 samples. The MCMC sample was based on a run of 100 million generations sampled every 10,000 steps with the first 10 million generations regarded as burn-in. We used the HKY + G model of nucleotide substitution without partitioning the coding region. A strict clock was used and the prior substitution rate was set to $1.691 \times 10^{-8}$ subs/site/year[76].

## Integrating interdisciplinary data and evidence

To depict the global picture of Kra-Dai language divergence and dispersal, we put the linguistic, archaeological, genetic, paleoecologic, and paleoclimatic data together to align this cross-disciplinary evidence in the temporal domain (Supplementary Data 13). For linguistic data, we inferred the divergence tempo of the Kra-Dai language by

counting the number of internal nodes of the language phylogeny. We used a sliding window approach with a length of 500 years and a shift step of 50 years. The tempo curve was smoothed by Local regression using weighted linear least squares and a 1st-degree polynomial model. It was implemented as the *smooth* function in MATLAB 2020b and the function parameter 'smooth span' was set to 0.1[77]. For archaeological data, we choose the data of archaeological sites in South China (including Zhejiang, Fujian, Hunan, Guangdong, and Yunnan) drawn from Hosner et al. [78]. The period was from 6000 to 2000 years BP. We counted the number of archaeological sites whose time range covered a specific time point of the year. And then the same smoothing approach was applied to obtain the trend curve. For genetic data, we used our Bayesian skyline plot reconstructed upon mtDNA data without any smoothing process. For paleoecologic data, we used the data derived from Gutaker et al.'s work in which the percent probability of tropical rice was displayed in the thermal niche (assuming a requirement of 2900 growing degree days at 10 °C bases) over time[39]. For paleoclimatic data, we used the Chinese Holocene Temperature Series which was synthetically reconstructed based on about 1397 temperature records during the Holocene in China[41,42].

### Reporting summary

Further information on research design is available in the Nature Portfolio Reporting Summary linked to this article.

## Data availability

Data are available through Supplementary Data Files. Supplementary Data 1: The lexical cognate database of Kra-Dai languages. Sheet 1 "Lexical items" shows the language entries and lexical items of 100 Kra-Dai languages. Sheet 2 "Binary coded sets" shows the binary coded form based on Sheet 1. Sheet 3 "Note" shows the reason for deleting specific lexical items and descriptions for the data. Supplementary Data 2: Node constraints with known historical information used to calibrate the divergence time calculations in BEAST. Supplementary Data 3: Model comparison among different combinations of models in BEAST. Sheet 1 "HME" shows the results compared by Harmonic Mean Estimator. Sheet 2 "Path Sampling" shows the results compared by the path sampling method. Supplementary Data 4: Possibilities of ancestral area. Supplementary Data 5: Comparison of time depth and root probability among versions of different settings. Supplementary Data 6: Delta scores and Q-residual scores. Supplementary Data 7: Information on archaeological sites in South China and MSEA. Supplementary Data 8: Resources of linguistic data. Supplementary Data 9: BEAST xml files, nexus files, MCC tree file, and the log file of the best-fitting model. Supplementary Data 10: Statistical significance among models of dispersal routes in the posterior samples of 3 independent runs of the RJMCMC. Supplementary Data 11: The control file, input file, and geographical distribution file used in the *BayesTraits* program. Three log files of MCMC runs were also included. Supplementary Data 12: The details of Kra-Dai population samples included in this study, including references, haplogroups, accession codes, etc. Supplementary Data 13: The interdisciplinary data and code used to plot Fig. 3. The time ranges and geographic locations of the archaeological sites were from the study of Hosner et al. [78]. (URL: https://doi.org/10.1594/PANGAEA.860072). The palaeoecological data were from the study of Gutaker et al. [39]. (URL: https://doi.org/10.1038/s41477-020-0659-6). The paleoclimatic data were from the study of Fang and Hou[41] (https://doi.org/10.13249/j.cnki.sgs.2011.04.013).

The 41 complete mitochondrial DNA sequences (fasta format) of Zhuang and Dong populations first reported in this study have been deposited in OMIX (https://ngdc.cncb.ac.cn/omix: accession no. OMIX002209) and the corresponding raw sequencing data in the Genome Sequence Archive for Human (https://ngdc.cncb.ac.cn/gsa-human: accession no. HRA005696) at the National Genomics Data Center, Beijing Institute of Genomics, Chinese Academy of Sciences,

following the regulations of the Human Genetic Resources Administration of China (HGRAC). In compliance with the regulations of the Ministry of Science and Technology of the People's Republic of China, the complete mitochondrial DNA sequences and the raw sequencing data contain information unique to an individual and thus require controlled access. Further analysis of these complete mitochondrial DNA sequences and sequencing data will be made available for collaborating researchers upon request, in compliance with the HGRAC's approval. For other public genetic data used in this study, the accession codes derived from previous published sources were listed in Supplementary Data 12.

## Code availability

Codes are available through Supplementary Data Files.

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

## Acknowledgements

We thank Xiuqi Fang and Guangliang Hou for providing the paleo-climatic data. We also thank Shi Yan's contribution to collecting genetic data previously. This research is supported by the National Natural Science Foundation of China (T2122007, 32070577, 32270670 and 31271338), National key research and development program (2020YFE0201600), National Social Science Foundation (20&ZD301), Shanghai Municipal Science and Technology Major Project (2017SHZDZX01), and the European Research Council (ERC) under the European Union's Horizon 2020 research and innovation programme (Grant Agreement No. 883700 TRAM). This work is also sponsored by "Shuguang Program" supported by Shanghai Education Development Foundation and Shanghai Municipal Education Commission (20SG06) and supported by the Guangxi Scholarship Fund of Guangxi Education Department and the Key Innovation Group of Digital Humanities Resource and Research at Shanghai Normal University.

## Author contributions

M.Z., Y.W. and H.Z. designed the research. Y.W., J.G., W.P. and M.Z. assembled and collated the linguistic and geographical data of the Kra-Dai languages. W.W., Q.B. and C.F. provided first-hand language documents from their previous linguistic fieldwork for some Kra-Dai languages. Y.P. and P.S. assembled and collated the archaeological data. H.Z. assembled the genetic data for the Kra-Dai people and performed the genetic analysis. Y.T. and M.Z. performed the linguistic analyses and interdisciplinary alignment. W.W., J.G., Y.P., P.S., Y.T., L.J., H.Z. and M.Z. discussed the results. Y.T., H.Z. and M.Z wrote the paper.

## Competing interests

The authors declare no competing interests.
