## [Peer Review File · Nature Communications]

REVIEWER COMMENTS

Reviewer #1 (Remarks to the Author):

Tao et al. applied the Bayesian phylogenetic methods to analyze the linguistic dataset containing 653 root-meaning traits compiled for 100 Kra-Dai languages. Coupled with genetic (mitochondrial DNA), archaeological, paleoecologic, and paleoclimatic data, the authors suggested that the Kra-Dai language diversification could coincide with their demic diffusion and agricultural spread shaped by the global climate change in the late Holocene.

In general, it's a lovely paper that publishes many new data on the Kra-Dai languages. The statistical methods are generally valid and correctly applied. The reference list nicely covers the relevant literature. In addition, The English writing is of sufficient quality. I would like to see a publication of it.

I also have the following points that will need the authors to take into consideration in their revision:

(1) I would suggest the manuscript can be structured into Introduction, Results, and Discussion sections to guide the readers to follow the context better.

(2) I am very glad to see the authors have integrated the genetic evidence in interpreting the results. Kra-Dai populations are suggested to have extensive genetic admixture with surrounding Han Chinese, Hmong-Mien and Tibeto-Burman speaking groups, showing frequent communication among those populations. Are there any language borrowings in those populations? Does the genetic admixture and language borrowing affect the phylogenetic topology of Kra-Dai?

The following genetic papers for your information:

Bin, X., Wang, R., Huang, Y., Wei, R., Zhu, K., Yang, X., Ma, H., He, G., Guo, J., Zhao, J., Yang, M., Chen, J., Zhang, X., Tao, L., Liu, Y., Huang, X., & Wang, C. C. (2021). Genomic Insight Into the Population Structure and Admixture History of Tai-Kadai-Speaking Sui People in Southwest China. *Frontiers in genetics*, 12, 735084.

Chen, J., Zhu, K., Yang, X., Wang, R., Ma, H., Tao, L., Liu, Y., Shen, Q., Yang, W., ... Huang, J. (2022). Fine-Scale Population Admixture Landscape of Tai-Kadai-Speaking Maonan in Southwest China Inferred From Genome-Wide SNP Data. *Frontiers in genetics*, 13, 815285.

He, G., Wang, Z., Guo, J., Wang, M., Zou, X., Tang, R., Liu, J., Zhang, H., Li, Y., Hu, R., Wei, L. H., Chen, G., Wang, C. C., & Hou, Y. (2020). Inferring the population history of Tai-Kadai-speaking people and southernmost Han Chinese on Hainan Island by genome-wide array genotyping. *European Journal of Human Genetics*, 28(8), 1111–1123.

Ren Z, Yang M, Jin X, Wang Q, Liu Y, Zhang H, Ji J, Wang C-C and Huang J (2022) Genetic substructure of Guizhou Tai-Kadai-speaking people inferred from genome-wide single nucleotide polymorphisms data. *Front. Ecol. Evol.* 10:995783.

Wang, M. G. et al. Reconstructing the genetic admixture history of Tai-Kadai and Sinitic people: Insights from genome-wide SNP data from South China. *Journal of Systematics and Evolution* (2021).

Huang, X. et al. Genomic Insights Into the Demographic History of the Southern Chinese. *Frontiers in Ecology and Evolution*, 556 (2022)

(3) The relationship between Kra-Dai populations and Bai Yue nationalities could be discussed more in this study. And the recent advances in genomic studies about the connection between the modern Kra-Dai populations and ancient southern East Asian ancestries can be addressed such as Chen et al.'s work in 2022.

Reference:

Chen H, Lin R, Lu Y, Zhang R, Gao Y, He Y, Xu S. Tracing Bai-Yue Ancestry in Aboriginal Li People on Hainan Island. *Molecular Biology and Evolution*. 2022, 39(10).

(4) minor typo: please check the abbreviation such as AN = Austronesian on Page 6.

Reviewer #2 (Remarks to the Author):

The manuscript proposes a hypothesis of Kra-Dai phylogeny by applying phylogenetic methods on a set of basic vocabulary from an impressive number of KD languages. However, due to methodological shortcomings and results that are not consistent with known linguistic facts, I cannot recommend this paper for evaluation.

What are the noteworthy results?

The noteworthy results are that

- The dates estimation of major diversifications is 4,000BP and 2,300BP. This seems plausible. The latter seems consistent with the what we know from linguistic contact and historical records. However, historical records also seems to suggest another major diversification event, i.e. spread of Southwestern Tai, around 900BP, but this is not detected in the study.
- Kra is a primary branch sister to one that includes the rest of the family. This is very interesting and possible but given the problematic results to be discussed later it is not clear if it is tenable.

Will the work be of significance to the field and related fields? How does it compare to the established literature? If the work is not original, please provide relevant references.

The work is far from convincingly demonstrating relationships among KD languages, let alone demonstrating historical and prehistorical population and language dynamics. But it is the first study that carry out the task of applying Bayesian methods to KD language data that covers the whole language family. It is also original in its interdisciplinarity, something that is much needed in the study of KD languages and populations that speak it. However, the study fails to critically assess their results with existing linguistic literature. The comparison was done only superficially.

Does the work support the conclusions and claims, or is additional evidence needed?

It is hard to say whether the work support the conclusions and claims. While the conclusion is valuable as a hypothesis, problems in the methodology and consequently the results cast doubt on the conclusion.

Are there any flaws in the data analysis, interpretation and conclusions? Do these prohibit publication or require revision?

The data analysis, interpretation and conclusions are consistent with each other. However, the results cast doubts on the methodology. More specifically, as the authors mention in the supplementary discussion, Saek belongs to the Northern Tai branch of Tai but it is grouped with Southwestern Tai languages, explaining that it may be due to lexical replacements. This result seriously cast doubt on the validity of the methods and the dataset. If we look more carefully at the tree, we will see that many Tai languages are misclassified. For example, both TswThaiTrang and TswBangkok are very similar dialects of Thai but are grouped in different branches. In addition, Shan varieties, i.e. TswAiton TswHsipaw, TswTaunggyi, TswMangshi, and TswMenglian, are dispersed all over. Though officially labeled as "Dai" in China, TswYuanjiang is in fact not a SWT language but a CT language.

Is the methodology sound? Does the work meet the expected standards in your field?

The application of the Bayesian methods meet the expected standards but this study ignores literature that criticizes against applying it to linguistic data, e.g. Pereltsvaig and Lewis (2015). While Bayesian methods have proven to be powerful and have become normal in historical linguistics, their validity has never been proven. However, my concerns has to do with the dataset rather than the computational methods.

- It is not clear how the data are coded. Table 1 contains many characters but no definition of the characters are given. So what are the "root-meanings" for the six characters with the concept "belly" for example.
- The author says that they manually labeled the forms in the sampled language on the basis of their linguistic knowledge. It is not clear how this was done.
- Why are loanwords removed. They are not different from any other kinds of lexical innovations.
- Is the tree rooted? The author didn't seem to have included as outgroup. Why not? How would that affect the analysis?

Is there enough detail provided in the methods for the work to be reproduced?

No. See the methodological problems raised above.

I have also included two files with specific comments. I hope you find them useful.

RESPONSE TO REVIEWERS' COMMENTS

We would like to express our sincere thanks to the referees for all the comments and suggestions. We have addressed the comments raised by the referees, and revised the manuscript entirely. Here are four prominent revisions in our manuscript:

1. We rearranged the structure of our manuscript following the sections of Introduction, Results, and Discussion to improve the readability.
2. We added a more comprehensive discussion for Kra-Dai language phylogeny and interdisciplinary evidence, which would be helpful to further understand the history of Kra-Dai languages.
3. We supplemented the content to elucidate how we processed our lexical data, and provided some examples to help readers to catch a glimpse of the comparative methods in historical linguistics.
4. We conducted more computational analyses such as using the four-point analysis to examine our results of Bayesian phylogenetic method. We considered dialectically that such methods were not flawless but remained powerful and promising to solve some traditional questions in historical linguistics like language diversification.

Additionally, point by point responses to the referees' comments are listed below this letter.

We hope that the revised version of the manuscript is now acceptable for publication in *Nature Communications*.

I look forward to hearing from you soon.

With best wishes,

Yours sincerely,

Menghan Zhang (Hanson), Ph.D.
Professor, Fudan University

REVIEWER COMMENTS

Reviewer #1 (Remarks to the Author):

Tao et al. applied the Bayesian phylogenetic methods to analyze the linguistic dataset containing 653 root-meaning traits compiled for 100 Kra-Dai languages. Coupled with genetic (mitochondrial DNA), archaeological, paleoecologic, and paleoclimatic data, the authors suggested that the Kra-Dai language diversification could coincide with their demic diffusion and agricultural spread shaped by the global climate change in the late Holocene.

In general, it's a lovely paper that publishes many new data on the Kra-Dai languages. The statistical methods are generally valid and correctly applied. The reference list nicely covers the relevant literature. In addition, The English writing is of sufficient quality. I would like to see a publication of it.

Response: Thank you.

I also have the following points that will need the authors to take into consideration in their revision: (1) I would suggest the manuscript can be structured into Introduction, Results, and Discussion sections to guide the readers to follow the context better.

Response: Thank you for your suggestions. We revised the structure of the manuscript following the sections of Introduction, Results, and Discussion.

(2) I am very glad to see the authors have integrated the genetic evidence in interpreting the results. Kra-Dai populations are suggested to have extensive genetic admixture with surrounding Han Chinese, Hmong-Mien and Tibeto-Burman speaking groups, showing frequent communication among those populations. Are there any language borrowings in those populations? Does the genetic admixture and language borrowing affect the phylogenetic topology of Kra-Dai?

The following genetic papers for your information:

Bin, X., Wang, R., Huang, Y., Wei, R., Zhu, K., Yang, X., Ma, H., He, G., Guo, J., Zhao, J., Yang, M., Chen, J., Zhang, X., Tao, L., Liu, Y., Huang, X., & Wang, C. C. (2021). Genomic Insight Into the Population Structure and Admixture History of Tai-Kadai-Speaking Sui People in Southwest China. *Frontiers in genetics*, 12, 735084.

Chen, J., Zhu, K., Yang, X., Wang, R., Ma, H., Tao, L., Liu, Y., Shen, Q., Yang, W., ... Huang, J. (2022). Fine-Scale Population Admixture Landscape of Tai-Kadai-Speaking Maonan in Southwest China Inferred From Genome-Wide SNP Data. *Frontiers in genetics*, 13, 815285.

He, G., Wang, Z., Guo, J., Wang, M., Zou, X., Tang, R., Liu, J., Zhang, H., Li, Y., Hu, R., Wei, L. H., Chen, G., Wang, C. C., & Hou, Y. (2020). Inferring the population history of Tai-Kadai-speaking people and southernmost Han Chinese on Hainan Island by genome-wide array genotyping. *European Journal of Human Genetics*, 28(8), 1111–1123.

Ren Z, Yang M, Jin X, Wang Q, Liu Y, Zhang H, Ji J, Wang C-C and Huang J (2022) Genetic substructure of Guizhou Tai-Kadai-speaking people inferred from genome-wide single nucleotide polymorphisms data. *Front. Ecol. Evol.* 10:995783.

Wang, M. G. et al. Reconstructing the genetic admixture history of Tai-Kadai and Sinitic people: Insights from genome-wide SNP data from South China. *Journal of Systematics and Evolution* (2021).

Huang, X. et al. Genomic Insights Into the Demographic History of the Southern Chinese. *Frontiers in Ecology and Evolution*, 556 (2022)

Response: We appreciate the reviewer for pointing this topic out. We thus added more discussions on population admixture and potential language borrowings of the KD-speaking population in the section “*The genetic and cultural admixture patterns of KD and surrounding populations*” in the Supplementary Information. The text added is as follows:

“The history of KD-speaking populations and their language culture is far from clear. Since KD populations lived at the crossroads where five main language families have spread and diversified¹²⁶, KD populations could not be simply modeled as inheriting directly from Bai Yue lineage and culture.

In South China, KD populations have experienced substantial contact with the aboriginal Hmong-Mien populations deeply from the beginning of the late Holocene^{98,127}. Specifically, for several geographically close ethnic groups of two language families, extensive genetic admixture was observed, and no clear genetic barrier existed among them^{128,129}. These admixture scenarios formed a “Hmong-Mien Cline” showing that Hmong-Mien-speaking individuals from west to east roughly have a decreased proportion of Hmong-Mien-related ancestry component and an increased proportion of KD-related ancestry component¹³⁰. The Tibeto-Burman populations were another ethnic group that came to South China in the late Holocene. In contrast to Hmong-Mien people, however, Tibeto-Burman people contributed little genetic influence to KD people^{127,128}. Moreover, Sinitic populations created powerful empires and expanded their political and military influences to South China in the last 2,000 years⁹⁷. Their dominant political power has greatly contributed to the population admixture in South China^{111,127}. In addition, Sinitic people also dominated cultural admixture and shift. These could be reflected in the presence of loanwords⁵¹, variations in phonetic structures³⁴, and grammatical system². Therefore, the admixture history of KD-speaking populations in South China was related to frequent gene flow and their cultural communications with surrounding people, especially Hmong-Mien and Sinitic populations.

In MSEA, Austroasiatic-speaking populations were the main local people whereas KD-speaking populations expanded to this region in the last 2,000 years^{123,126}. The genetic study revealed that heterogeneity in admixture with local Austroasiatic groups and geographic proximity primarily shaped the genetic structure of KD people¹³¹. In addition, Sino-Tibetan, Hmong-Mien, and Austronesian groups which also migrated to MSEA contributed limited genetic ancestry to KD people^{126,131}. Therefore, the extensive contact between the groups of different language families resulted in cultural diffusion and even cultural shift in MSEA¹³².

In summary, extensive contact with surrounding populations collectively shaped present-day KD-speaking populations and their languages. Meanwhile, we expected more detailed studies to shed further light on their complex history.”

(3) The relationship between Kra-Dai populations and Bai Yue nationalities could be discussed more in this study. And the recent advances in genomic studies about the connection between the modern Kra-Dai populations and ancient southern East Asian ancestries can be addressed such as Chen et al.’s work in 2022.

Reference:

Chen H, Lin R, Lu Y, Zhang R, Gao Y, He Y, Xu S. Tracing Bai-Yue Ancestry in Aboriginal Li People on Hainan Island. *Molecular Biology and Evolution*. 2022, 39(10).

Response: We added more discussions on the relationship between Kra-Dai populations and Bai Yue nationalities from Line 542 to 549 in the section “The prehistoric cultures in coastal Southeast China ~ 5,000 years ago” in the Supplementary Information. The text added is as follows:

“In ancient China, the term “Bai Yue” referred to the “hundreds of tribes,” which were collectively known as ancient indigenous KD-speaking populations living in present-day coastal southeast China^{106,110}. A recent genetic study confirmed that Bai Yue ancestry was widely distributed in KD-speaking populations in South China and MSEA.¹¹¹. Their study found that although other genetic components were present, the Bai Yue lineage was dominant in contemporary KD-speaking populations. In other words, the diverse present-day KD-speaking populations descended from their common ancestor, the ancient Bai Yue lineage, which underwent different migration, admixture, and isolation over time.”

(4) minor typo: please check the abbreviation such as AN = Austronesian on Page 6.

Response: We thank the reviewer for pointing this out. Accordingly, we collated the abbreviation in the revised manuscript.

Reviewer #2 (Remarks to the Author):

The manuscript proposes a hypothesis of Kra-Dai phylogeny by applying phylogenetic methods on a set of basic vocabulary from an impressive number of KD languages. However, due to methodological shortcomings and results that are not consistent with known linguistic facts, I cannot recommend this paper for evaluation.

(1) What are the noteworthy results?

- The noteworthy results are that - The dates estimation of major diversifications is 4,000BP and 2,300BP. This seems plausible. The latter seems consistent with the what we know from linguistic contact and historical records. However, historical records also seems to suggest another major diversification event, i.e. spread of Southwestern Tai, around 900BP, but this is not detected in the study.

Response: One goal of our study was to align the interdisciplinary data to depict the global prehistorical picture of the KD language divergence and dispersal in South China and Mainland Southeast Asia. Based on this interdisciplinary alignment, we could further investigate the possible driving forces of KD language evolution such as environmental change, agricultural development, and potential demographic activities of KD-speaking people. However, limited by the temporal range of archeological data, we could not align the dynamic change of archeological site numbers to other disciplinary evidence in the past 2000 years. Therefore, we discussed surrounding the two major diversifications at 4,000 BP and 2,300 BP. In the original Supplementary Information, we mentioned briefly that the historical events at 900 BP could also be critical to the evolution of KD languages.

It is no doubt that the spread of Southwestern Tai languages should be an important event during the evolutionary process of KD languages. We indeed observed this event according to the rate shift of language diversification at about 900 years BP. The observation could be revealed using the first-order differential curve (Figure R1) of the language diversification originally in Figure 3a. We added the corresponding figure and discussions in the section “*The history of the KD-speaking populations in the past 4,000 years*” in the revised Supplementary Information, as follows:

“For example, during the Qin Dynasty, a war was waged to conquer Bai Yue after unifying six states (around 2,200 years BP)². The war forced many Bai Yue populations to migrate, which later resulted in the increasing divergence rate (interval 1 in Figure R1) and the divergence of Kam-Tai languages (around 1,950 years BP). Subsequently, the second migration wave occurred in the Tang Dynasty, when the Zhuang ancestors resisted the reign of Tang Dynasty for nearly one hundred years (around 1,200 years BP)². However, the rebels of Zhuang ancestors were suppressed and could only migrate to MSEA, resulting in a rapid divergence in this period (interval 2 in Figure R1). This was followed by the divergence of the Southwestern Tai languages and their sister group of Central Tai languages (around 1,179 years BP). During the Song Dynasty (around 950 years BP)², the third migration occurred as a result of the failure of the uprising led by minority leader Nungz Cigauh. This led to another large-scale migration southwestward into mainland Southeast Asia, which coincided with a rapid increase in the language divergence rate (interval 3 in Figure R1) and the divergence of Southwestern Tai languages (around 824 years BP). This migration might have contributed to the unification of Thailand in the following decades. Our analysis, combined with reliable historical documents, suggested that political power was a major influence on the demographic activities of the KD people during the second period.”

Figure R1. The variation rate of the number of nodes of KD phylogeny. The first-order differential curve of the language diversification originally in Figure 3a. The three intervals marked by dark bands in the 2nd period were three important historical events that related to KD-speaking populations, respectively.

(2) - Kra is a primary branch sister to one that includes the rest of the family. This is very interesting and possible but given the problematic results to be discussed later it is not clear if it is tenable.

Response: The observation that the Kra language was a primary branch sister to one that includes the rest of the KD family, was consistent with the findings of previous historical linguistic studies (Liang and Zhang, 1996; Edmondson and Solnit, 1997; Diller *et al.*, 2008; Chamberlin, 2016; Li, 2019).

Different from Bayesian phylogenetic reconstruction for Indo-European (Boukeart *et al.*, 2012) and Austronesian languages (Gray *et al.*, 2009), it was noted that we did NOT set any monophyletic constraint on each well-known branch such as Ong-be, Hlai, Tai. And we did NOT set the global typological structure as a prior setting as well. These model settings were primarily important to the Bayesian phylogenetic reconstruction in our study, which was resistant to potential subjective assumptions on the phylogeny. Therefore, our phylogenetic tree of Kra-Dai languages was a posterior result affected by the Birth Death Skyline (BDSKY) Contemporary BDSParam model we used and the lexical cognate data. In particular, the reason for choosing the BDSKY Contemporary BDSParam model was that the language samples were all sampled at the same point in time. Moreover, this model can account for incomplete sampling of languages to give better estimates of the tree topology and timing (Hoffman *et al.*, 2021). Therefore, the model integrated the uncertainties on the incomplete sampling of languages. To estimate the divergence time, we followed our previous work on the Sino-Tibetan languages (Zhang *et al.*, 2019), we adopted several time calibrations of internal nodes supported by chronological estimations in linguistic and ethnological literature but not following genetic and archaeological evidence (see Table S2. Calibrations on Tree).

For linguistic data, in addition, we compiled the lexical items following the Swadesh 100-word list and their corresponding IPA transcriptions from several linguistic publications available and the first-hand language field works proceeded by the co-authors. The cognates were identified based on linguistic criteria of regular sound correspondences and then collated by seven independent linguistic scholars (i.e., Yuanchen Wei, Jiaqi Ge, Changzhong Fu, Wenmin Wang, Qianqi Bi, and Wuyun Pan) several times under the framework of historical comparative method in linguistics.

In summary, we consider that our data and evolutionary model should be reliable, and accordingly our results should be tenable and reproducible.

References

1. Liang, M. & Zhang, J. R. *An Introduction to the Kam-Tai Languages (Chinese version)*. (China Social Sciences Press, 1996).
2. Edmondson, J. A. & Solnit, D. B. *Comparative Kadai: the Tai branch*. (Summer Institute of Linguistics Publications in Linguistics, 1997).
3. Diller, A., Edmondson, J. & Luo, Y. *The Tai-Kadai Languages*. (Routledge, 2008).
4. Chamberlain, J. R. Kra-Dai and the proto-history of South China and Vietnam. *The Journal of the Siam Society* **104**, 27-77 (2016).
5. Li, P. J.-k. Dongdai yuzu de zujudi, kuosan ji qi shiqianwenhua (Chinese version). *Journal of East Linguistics* (2019).
6. Bouckaert, R. *et al.* Mapping the origins and expansion of the Indo-European language family. *Science* **337**, 957-960 (2012).
7. Gray, R. D., Drummond, A. J. & Greenhill, S. J. Language phylogenies reveal expansion pulses and pauses in Pacific settlement. *Science* **323**, 479-483 (2009).
8. Hoffmann, K., Bouckaert, R., Greenhill, S. J. & Kühnert, D. Bayesian phylogenetic analysis of linguistic data using BEAST. *Journal of Language Evolution* **6**, 119-135 (2021).
9. Zhang, M., Zheng, H. X., Yan, S. & Jin, L. Reconciling the father tongue and mother tongue hypotheses in Indo-European populations. *Natl Sci Rev* **6**, 293-300 (2019).

(3) Will the work be of significance to the field and related fields? How does it compare to the established literature? If the work is not original, please provide relevant references.

The work is far from convincingly demonstrating relationships among KD languages, let alone demonstrating historical and prehistorical population and language dynamics. But it is the first study that carry out the task of applying Bayesian methods to KD language data that covers the whole language family. It is also original in its interdisciplinarity, something that is much needed in the study of KD languages and populations that speak it. However, the study fails to critically assess their results with existing linguistic literature. The comparison was done only superficially.

Response: For the linguistic aspect, our work aimed to examine some controversies on the linguistic relationships among five well-described language branches: Kra, Hlai, Ong-be, Tai, and Kam-Sui. Accordingly, we established the largest lexical cognate database for KD languages, constructed their Bayesian phylogenetic tree, then estimated the divergence time, and further inferred the potential dispersal center under the data-driven computational framework. It was noted that the previous works for the classification of the whole KD language phylum were based on the linguistic evidence of both lexical and phonological innovations but had never been attested by computational methods (Sidewell and Jenny, 2021: 229). The Bayesian phylogeny of KD languages we obtained showed

high posterior possibility values for each branch, indicating that all these five branches should be monophyletic. Also, we obtained relatively strong support for the relationships among these five branches (see Figure 1 and Figure S4. Densitree). Our phylogenetic results supported the traditional historical linguist Ostapirat's classification (Ostapirat, 2017).

Given the dated language phylogeny, we further integrated the linguistic evidence with the other disciplinary evidence on the spatiotemporal domain. One significance of our interdisciplinary study was to demonstrate the macroevolutionary picture of KD language divergence and dispersal using quantitative approaches. In contrast to previous studies of Indo-European (Bouckaert *et al.*, 2012), Austronesian (Gray *et al.*, 2009), and other language families which adopted the research evidence to explain the linguistic observations, our study integrated multi-disciplinary data and aligned them at temporal domain primarily to investigate the driving forces on the KD language evolution along with the demic activities of KD-speaking populations and other ethnic populations in South China and MSEA. The interdisciplinary alignment adopted in our study should be a critical approach (e.g., Robbeets *et al.*, 2021) to deeply understand the evolutionary mechanisms underlying the language diversity we observed in the world, especially in East and Southeast Asia. In sections “*The prehistoric cultures in coastal Southeast China ~ 5,000 years ago*” and “*The history of the KD-speaking populations in the past 4,000 years*” of the revised version of Supplementary Information, we supplemented and demonstrated the prehistorical and historical population and language dynamics based on interdisciplinary evidence.

According to the reviewer's suggestions, we added the comparison of our results and previous linguistics' classification in each of the five language branches in the section “*Phylogenetic topology of the KD languages*” in the revised version of Supplementary Information, as follows:

“In contrast to the high-level relationships, the low-level branches showed a more complicated phylogenetic relationship that was not completely consistent with traditional linguists' expectations. In particular, the Kra languages majorly conformed to Ostapirat's view²¹ which classified Buyang languages and Pubiao in a monophyletic group; while Gelao languages, Lachi and Laha in another. However, the Paha language (KraBuyangBH) was estimated to be more related to Lachi and Laha languages in our results, whereas Ostapirat suggested that the Paha language should be related to the Buyang-Pubiao group. The Hlai languages majorly conformed to Norquest's classification¹⁹, which suggested that Cun and Nadou languages should be a sister group to Meifu dialects (HlaiChangjiang, HlaiXifang) and supported a monophyletic group of Qi dialects (HlaiBaoting, HlaiQiandui, and HlaiTongza). However, Run (Bendi) dialects (HlaiBaisha and HlaiYuanmen) were placed as a sister group of Qi dialects in our results but not a sister group of Meifu dialects in linguists' view. The Ong-Be languages were completely consistent with Ostapirat's view⁶², which suggested that the Jizhao dialect branched first and the other Ong-Be languages were split into western and eastern groups. In contrast, in the Kam-Sui branch, we could not confidently determine the relationships among Mulam, Then, Mak, Jin, Maonan, and Chadong languages due to the low posterior values of the internal nodes. However, the place of Biao and Lakkia in our result supported Solnit's view⁶³ regarding Biao and Lakkia as a monophyletic group coordinated with Kam-Sui. Finally, the Tai languages were split into two parts roughly based on their locations. The Northern Tai languages were grouped with Yongnan dialects of Central Tai languages (TcFusui, TcShangsi, TcLongAn, TcQin Zhou, and TcYongning), whereas the Southwestern Tai languages were grouped with the other Central Tai languages. The inexistence of monophyletic Central Tai languages was advocated by Edmondson^{50,61} and Pittayaporn^{6,7}. In addition, we found that six Shan varieties

(TswAiton TswHsipaw, TswTaunggyi, TswMangshi, TswMenglian, and TswKhuen)⁶⁴⁻⁶⁷ collectively consisted of a paraphyletic group, a lower-level clade of Southwestern Tai languages. Despite other non-Shan varieties included in this clade, this node implied that a Proto-Shan language might yet be present. However, note that the low posterior value of this node (= 0.31) suggested that this internal node should not be robust in our results. In addition, the Saek language as a Northern Tai language seemed to be misplaced into the Southwestern Tai languages supported by a posterior value of 0.52.

There were some reasons for such misplacements observed in our phylogenetic tree. First, to give a straightforward display, we presented the maximum clade credibility (MCC) tree based on the posterior samples calculated from the MCMC method. The MCC tree could be interpreted as a global optimum tree for clade credibility. For a given node, lower posterior value represented less stable structure. Thus, we could not avoid the misplacements that might occur in several clades. Second, our work was solely relying on lexical cognates while most traditional linguistic classifications were based on both phonology and morphology. Third, the Swadesh 100-word list could not provide sufficient resolutions to distinguish the low-level relationships among KD languages, especially when these languages experienced rapid differentiation in a short period and substantial language contacts (e.g., lexical borrowings) especially occurring during the period of initial language divergence. Fourth, the borrowing-prone languages were difficult to evaluate their linguistic relatedness, because it would be difficult to determine which linguistic traits were inherited from a common ancestor and which were borrowed from other languages.

Here, we took Saek as an example. Saek is a minority language of Northern Tai in Thailand but is substantial contact-induced change from its surrounding Southwestern Tai languages (e.g., Thai and Lao languages)^{68,69}. To explore which reason has led to the misplacement of Saek and address the authentic genealogical classification of Saek, Northern Tai, and Southwestern Tai, we performed the four-point analysis which could provide the possibilities of a specific two-to-two partition directly estimated from the linguistic data given the sub-tree structure. As this method examined all the cognate sets one by one, the possibility for a given structure could be considered to be the proportion of shared cognate sets (inheriting from a common ancestor or borrowing from other languages) in the tested language and its nearest language in the given structure⁷⁰. Following the computational procedure in our previous study⁷⁰, the results of the four-point analysis showed that the possibility for (Saek, Tsw)-(Tn, Others) was 0.3716, for (Saek, Tn)-(Tsw, Others) was 0.6275, and (Saek, Others)-(Tn, Tsw) was 9.2593×10^{-4} . The subtree structure for (Saek, Tn)-(Tsw, Others) was moderately supported, indicating that Saek should belong to the Northern Tai group rather than the Southwestern Tai. However, the subtree structure for (Saek, Tsw)-(Tn, Others) was weakly supported, indicating that Saek might have considerable borrowings (37.16%) from Southwestern Tai languages. Such a vast number exceeded the 20% limit of the Bayesian phylogenetic methods for borrowing words⁷¹, which could potentially twist the tree structure. Therefore, we suggested that the given lexical cognate data were sufficient to distinguish the fine-scale relationship among similar languages under given circumstances. The misplacement of Saek in Bayesian phylogenetic results was resulted from methodological inadaptability for distinguishing borrowing-prone languages. This was also supported by linguistic views that Saek could experience substantial borrowings or replacements from its surrounding Southwestern Tai languages^{68,69}.”

References

1. Sidwell, P. & Jenny, M. *The Languages and Linguistics of Mainland Southeast Asia: A Comprehensive Guide*. Vol. 8 (Walter de Gruyter GmbH & Co KG, 2021).
2. Ostapirat, W. Kra-Dai in Southern China (Invited talk in the Nankai University). (Nankai University, 2017).
3. Zhang, M., Zheng, H. X., Yan, S. & Jin, L. Reconciling the father tongue and mother tongue hypotheses in Indo-European populations. *Natl Sci Rev* **6**, 293-300 (2019).
4. Greenhill, S. J. *et al.* Evolutionary dynamics of language systems. *Proceedings of the National Academy of Sciences* **114**, E8822-E8829 (2017).
5. Thomason, S. G. & Kaufman, T. *Language contact*. (Edinburgh University Press, 2001). Atkinson, Q. D. & Gray, R. D.
6. Bouckaert, R. R. & Heled, J. DensiTree 2: Seeing trees through the forest. *BioRxiv*, 012401 (2014).
7. Atkinson, Q. D. & Gray, R. D. Curious parallels and curious connections—phylogenetic thinking in biology and historical linguistics. *Systematic biology* **54**, 513-526 (2005).
8. Gray, R. D., Drummond, A. J. & Greenhill, S. J. Language phylogenies reveal expansion pulses and pauses in Pacific settlement. *Science* **323**, 479-483 (2009).
9. Greenhill, S. J. & Gray, R. D. Basic vocabulary and Bayesian phylo-linguistics: Issues of understanding and representation. *Diachronica* **29**, 523-537 (2012).
10. Neureiter, N. *et al.* Detecting contact in language trees: a Bayesian phylogenetic model with horizontal transfer. *Humanities and Social Sciences Communications* **9**, 1-14 (2022).
11. Koile, E., Greenhill, S. J., Blasi, D. E., Bouckaert, R. & Gray, R. D. Phylogeographic analysis of the Bantu language expansion supports a rainforest route. *Proceedings of the National Academy of Sciences* **119**, e2112853119 (2022).
12. Zhang, M., Yan, S., Pan, W. & Jin, L. Phylogenetic evidence for Sino-Tibetan origin in northern China in the Late Neolithic. *Nature* **569**, 112-115 (2019).
13. Robbeets, M. *et al.* Triangulation supports agricultural spread of the Transeurasian languages. *Nature* **599**, 616-621 (2021).

(4) Does the work support the conclusions and claims, or is additional evidence needed?

It is hard to say whether the work support the conclusions and claims. While the conclusion is valuable as a hypothesis, problems in the methodology and consequently the results cast doubt on the conclusion.

Response: To reconstruct KD language prehistory, we adopted the state-of-the-art Bayesian phylogenetic approach implemented in the BEAST program. Different from the controversial methods of lexicostatistics and glottochronology in historical linguistics, recent advances in Bayesian phylogenetic methods originally from evolutionary biology provide alternative opportunities to permit flexible evolutionary models (i.e., site model, clock model, and tree models). It has been widely used in reconstructing the prehistories of languages and is regarded as a powerful tool for inferring evolutionary tempo and mode of change in language families worldwide (Gray, *et al.*, 2009; Bouckaert, *et al.*, 2012 and 2018; Grollemund, *et al.*, 2015; Zhang, *et al.*, 2019).

Surrounding the topic of the methodology of phylogenetic reconstruction of languages, Greenhill *et al.* (2020) have pointed out several limitations of the traditional historical comparative method in constructing language family trees, “*traditional historical linguists do not use an explicit optimality criterion to select the best tree, nor do they use an efficient computer algorithm to search*

for the best tree.....Traditional language family trees suffer from two additional limitations: their branching patterns reflect only a relative chronology and they contain no information about how strong the support is for any proposed subgroupings.” In contrast, “Bayesian phylogenetic methods provide a useful supplement to the comparative method. They enable us to build trees with explicit estimates both of branch lengths and of subgrouping uncertainty in an objective, repeatable manner.”

Notably, Bayesian phylogenetic methods have also been challenged due to some deep problems with linguistic analysis. Accordingly, we discussed this issue in Comment 6 and in section “Bayesian phylolinguistics, proper-used or misused?” in the revised version of Supplementary Information.

References

1. Gray, R. D., Drummond, A. J. & Greenhill, S. J. Language phylogenies reveal expansion pulses and pauses in Pacific settlement. *Science* **323**, 479-483 (2009).
2. Bouckaert, R. *et al.* Mapping the origins and expansion of the Indo-European language family. *Science* **337**, 957-960 (2012).
3. Grollemund, R. *et al.* Bantu expansion shows that habitat alters the route and pace of human dispersals. *Proceedings of the National Academy of Sciences* **112**, 13296-13301 (2015).
4. Bouckaert, R. R., Bowern, C. & Atkinson, Q. D. The origin and expansion of Pama–Nyungan languages across Australia. *Nature ecology & evolution* **2**, 741-749 (2018).
5. Zhang, M., Yan, S., Pan, W. & Jin, L. Phylogenetic evidence for Sino-Tibetan origin in northern China in the Late Neolithic. *Nature* **569**, 112-115 (2019).
6. Greenhill, S. J., Heggarty, P. & Gray, R. D. in *The handbook of historical linguistics* Vol. 2 *Bayesian Phylolinguistics* Ch. 11, 226-253 (2020).

(5) Are there any flaws in the data analysis, interpretation and conclusions? Do these prohibit publication or require revision?

The data analysis, interpretation and conclusions are consistent with each other. However, the results cast doubts on the methodology. More specifically, as the authors mention in the supplementary discussion, Saek belongs to the Northern Tai branch of Tai but it is grouped with Southwestern Tai languages, explaining that it may be due to lexical replacements. This result seriously cast doubt on the validity of the methods and the dataset. If we look more carefully at the tree, we will see that many Tai languages are misclassified. For example, both TswThaiTrang and TswBangkok are very similar dialects of Thai but are grouped in different branches. In addition, Shan varieties, i.e. TswAiton TswHsipaw, TswTaunggyi, TswMangshi, and TswMenglian, are dispersed all over. Though officially labeled as “Dai” in China, TswYuanjiang is in fact not a SWT language but a CT language.

Response:

(a) To address the comment on the misplacement of the Saek language, we performed the four-point analysis to evaluate the reliability of language classification based on the inferred phylogenetic tree. The analysis can provide the possibilities of a specific two-to-two partition directly estimated from the linguistic data given the sub-tree structure. In this study, we evaluated the reliabilities of the relationship among Tsw, Saek, and Tn. As this method examined all the cognate sets one by one, the possibility for a given structure could be considered to be the proportion of shared cognate sets (inheriting from a common ancestor or borrowing from other languages) in the tested language and

its nearest language in the given structure. Following the computational procedure in our previous study (Zhang *et al.*, 2019), we showed the results of the four-point analysis that the possibility for (Saek, Tsw)-(Tn, Others) was 0.3716, for (Saek, Tn)-(Tsw, Others) was 0.6275, and (Saek, Others)-(Tn, Tsw) was 9.2593×10^{-4} . The subtree structure for (Saek, Tn)-(Tsw, Others) was moderately supported, indicating that Saek should belong to the Northern Tai branch of Tai rather than the Southwestern Tai language. However, the subtree structure for (Saek, Tsw)-(Tn, Others) was weakly supported, indicating that Saek could have undergone substantial borrowings from Southwestern Tai languages (Khanittanan & Yang, 2003 and 2004). Such vast borrowings (37.16%) exceeded the limit of the Bayesian phylogenetic method for borrowing words (20%) (Atkinson *et al.*, 2005; Gray *et al.*, 2009) and could twist the part of tree structure. However, since Saek was of great significance for linguistic studies, we did not exclude Saek from our dataset and discussed our results of Saek in the section “*Phylogenetic topology of the KD languages*” in the revised version of Supplementary Information. Despite that, compared with the settings which constrained Saek with Tn or excluded Saek from our data, respectively, we could still obtain consistent and robust conclusions for the linguistic relatedness of five language groups, time depth and dispersal center (see Comment 6, Figure R2, Figure R3, and Table R1).

(b) For the comment on the different places of TswThaiTrang and TswBangkok, we considered that different criteria of classification might lead to this misplacement. The traditional classification was based on phonological innovations, whereas our methods reconstructed the language relationship from the lexical view. We acknowledged that lexical evidence could not be the only criterion for language classification but remained crucial and necessary for reconstructing language trees (see more detailed discussion in Comment 6).

(c) For the comment on the absence of a monophyletic clade of varieties of Shan language, according to our revised version of the language tree, we could observe a paraphyletic group of Shan varieties, a lower-level clade of Southwestern Tai languages including TswAiton TswHsipaw, TswTaunggyi, TswMangshi, TswMenglian, and TswKhuen (Brown, 1985; He, 2003; Gogoi *et al.*, 2020). In addition, TswChiangmai, TswNgheAn, and TswJinghong were also included in this clade, although these languages were not considered to be Shan varieties. Notably, the posterior value of the node of “Proto-Shan” was only 0.31, indicating that this node was not robust in our results. We speculated that extensive cultural interactions and substantial lexical borrowings deeply affected Shan varieties and several non-Shan languages because Shan populations were an influential power in MSEA (He, 2003). In addition, the relationship among Shan varieties could be distinguished by phonological or morphological traits rather than Swadesh 100-word list. However, despite the misplacement of Shan varieties, the high-level internal nodes were still stable. As mentioned in Comment 6, we constrained the Shan varieties as a monophyletic group and performed the same computational analysis. Such treatment also gave the conclusions consistent with those of default settings (Figure R2, Figure R3, and Table R1). This suggested that, in our study, Bayesian phylogenetic methods were robust for inferring high-level structure and also needed urgent improvement to adapt them for distinguishing low-level branches accurately.

(d) For the comment on the classification of the Yuanjiang dialect, according to the widely-used linguistic database of Glottolog (Glottocode: taih1246) and Ethnologue (ISO 639-3 code: tiz), the

Yuanjiang dialect (also named as Tai Ya) is a Southwestern Tai language of southern China. And other linguistic literature such as Xing (1989) and Zhou & Luo (2001) also claim that the Yuanjiang dialect of Tai languages should belong to the Southwestern Tai clade. Accordingly, we considered the Yuanjiang dialect as a language of Southwestern Tai languages.

References:

1. Zhang, M., Yan, S., Pan, W. & Jin, L. Phylogenetic evidence for Sino-Tibetan origin in northern China in the Late Neolithic. *Nature* **569**, 112-115 (2019).
2. Khanittanan, W. & Yang, G. Saek Language (Part One) (Chinese version). *Nankai Linguistics*, 154-181+187 (2003).
3. Khanittanan, W. & Yang, G. Saek Language (Part Two) (Chinese version). *Nankai Linguistics*, 171-196 (2004).
4. Atkinson, Q. D. & Gray, R. D. Curious parallels and curious connections—phylogenetic thinking in biology and historical linguistics. *Systematic biology* **54**, 513-526 (2005).
5. Gray, R. D., Drummond, A. J. & Greenhill, S. J. Language phylogenies reveal expansion pulses and pauses in Pacific settlement. *Science* **323**, 479-483 (2009).
6. Brown, J. M. *From ancient Thai to modern dialects and other writings on historical Thai linguistics*. (White Lotus Company, 1985).
7. He, P. Miandian Fengjianwangchao Shili de Beikuo yu Danbang de Xingcheng (Chinese version). *South and Southeast Asian Studies*, 44-52 (2003).
8. Gogoi, P., Morey, S. & Pittayaporn, P. The Tai Ahom sound system as reflected by the texts recorded in the bark manuscripts. *Journal of the Southeast Asian Linguistics Society* **13**, 14-42 (2020).
9. Xing, G. W. Upper Hongjin Dai Ya Language. Language Publishing House (1989).
10. Zhou, Y. W. & Luo, M. Z. A Study of Dai Dialects. Ethnic Publishing House (2001).

(6) Is the methodology sound? Does the work meet the expected standards in your field?

The application of the Bayesian methods meet the expected standards but this study ignores literature that criticizes against applying it to linguistic data, e.g. Pereltsvaig and Lewis (2015). While Bayesian methods have proven to be powerful and have become normal in historical linguistics, their validity has never been proven. However, my concerns has to do with the dataset rather than the computational methods.

Response: According to Pereltsvaig and Lewis's book, "*The Indo-European Controversy: Facts and Fallacies in Historical Linguistics*" (Pereltsvaig and Lewis, 2015), several critical problems with linguistic analysis were pointed out specifically for the work of Bouckaert *et al.* in 2012. In this book, they proposed the **linguistic fallacies**, **dating problems**, and **historical-geographical failure** of the Bayesian phylogenetic analysis. It was noted that some problems were specific to Bouckaert *et al.*; whereas other problems were inevitable in any methods of reconstructing language family trees but would not affect our main results.

First, Pereltsvaig and Lewis demonstrated four specific linguistic fallacies, including (1) ignoring the differences between similarities that reflect shared retentions and those that reflect shared innovations; (2) examining only lexical material, which is intrinsically unreliable; (3) inadequately identifying borrowings, especially those occurring between closely related languages; (4) ignoring the misplacement of individual languages on the family tree.

For fallacy (1), in our study, we integrated the knowledge of both shared innovations and shared retentions to compile the list of regular sound correspondences. Based on this, we performed traditional comparative methods on our linguistic data to generate cognate sets. Then, these cognate sets were input for the calculation of Bayesian phylogenetic methods. Specifically, Bayesian phylogenetic methods excel at the quantitative estimation of language trees when various shared innovations are used. In other words, Bayesian phylogenetic methods are a powerful tool to supplement traditional linguistic scholarship, but not to replace it.

For fallacy (2), we must admit that our work was solely relying on lexical cognates. In general, different language subsystems should experience distinct evolutionary processes from the past to the present, such as different evolutionary patterns of phonology and lexicon (Zhang *et al.*, 2019), various evolutionary rates of linguistic features (Greenhill *et al.*, 2017), degrees of horizontal influence on phonological, grammatical, and lexical sub-systems (Thomason and Kaufman, 2001). Accordingly, all these processes would make different results of language classifications when we used various linguistic features to investigate language relationships. Compared with other linguistic features, the lexicon should be suitable for establishing relatedness because of availability, stability, comprehensiveness, and size advantage. Greenhill *et al.* (2020) pointed out that “*Lexical data are universally available (even if somewhat more limited in polysynthetic languages)*”. Moreover, the grammatical, phonological, and even phonetic traits might not be suitable for dating because they could vary more freely and faster than items of basic vocabulary (Greenhill *et al.*, 2017 and 2020). In addition, lexical materials were identified as cognate sets based on a comprehensive view including phonological and morphological knowledge, “*to define cognate sets relies on the comparative method, its sound-change laws, and reconstructions..... Cognacy assignment, when properly performed, integrates and rests on all of the data, methodology, and findings of orthodox comparative-historical linguistics, not least in phonology and morphology*” (Greenhill *et al.*, 2020). Compared with a large amount of cognacy in lexical meanings, Ringe *et al.* found few data characters in phonology and morphology to reconstruct higher-order nodes (Ringe *et al.*, 2002). They admitted, “*the worst news is yet to come: the vast majority of our well-behaved monomorphic characters simply define one or more of the ten uncontroversial subgroups of the family, contributing nothing to their higher-order subgrouping*”.

For fallacy (3), we admitted that the identification of borrowings is a fundamental work in our study. We endeavored to identify definite borrowings in KD languages as possible (details see our response in Comment 9) but we could not avoid that a small number of undetectable borrowings might exist. Fortunately, several previous studies of computational simulations on phylogenetic methods gave us more confidence that without any linguistic constraint on the phylogenetic reconstruction, the number of undetected borrowings should be very great (>20%) to substantially bias either the tree topology or the date estimates (Atkinson *et al.*, 2005; Gray *et al.*, 2009). More importantly, the tree structure emphasizes the vertical process of language diversification rather than their horizontal contacts and admixture. The extent of language contact could be measured by a delta score and Q residual value which was discussed in the section “*The homoplasy in the KD language phylogeny*” in Supplementary Information.

For fallacy (4), we should note that the phylogenetic tree just exhibits a hypothetical diagram of language diversification. The tree’s topological structure relies on what the input data are, and the topological uncertainties are shown by the posterior value of each internal node. The higher posterior values show more robust evidence for grouping the downstream languages as a

monophyletic group. As shown in our tree, the posterior values of higher-order internal nodes were almost higher than 0.9, which reliably supported the relationships of the five branches of KD languages. Indeed, we observed low posterior values for several lower-order internal nodes, which could be resulted from potential lexical borrowings or the limited resolution of lexical cognates for distinguishing languages. However, the minor misplacements of individual languages at the lower levels could not twist the high-level relationships among language branches, and not affect the overall shape of the phylogenetic tree (Greenhill and Gray, 2012). Accordingly, the Swadesh 100-word list could not provide sufficient resolutions to distinguish the low-level relationships among languages, especially when these languages experienced rapid differentiation in a short recent period and abundant language contacts (e.g., borrowings) during the period of initial language divergence. In other words, if we attempted to reconstruct a fine-scale topological structure at the low level of the phylogenetic tree, we should obtain more detailed and abundant linguistic materials. Moreover, the Gray-Atkinson team relied in part on Dyen's database but not Starostin's Tower of Babel database, and the two databases lead to different results (Pereltsvaig and Lewis, 2015: 89). In contrast, in our study, we integrated the databases of Starostin, Gedney, Liang & Zhang and many other linguistic documents derived from the first-hand language field surveys (see previous contents and Table S7. Data Resources). Accordingly, the bias of data selection in our study could be less than the work of Bouckaert *et al.* (2012). In other words, our dataset for KD languages would be more comprehensive.

In addition, the relationships among five branches of KD languages were exhibited by the maximum clade credibility (MCC) tree based on the posterior samples calculated from the MCMC method. The MCC tree could be interpreted as a global optimum tree. Thus, we could not exclude the possibilities that several internal nodes in the MCC tree would not be present in other posterior samples. These implied that the monophyletic groups of KD languages in the MCC tree could be paraphyletic ones in reality. These uncertainties were also shown by the DensiTree (Bouckaert and Heled, 2014) (see Figure S4. DensiTree). Collectively, these limitations could result in minor misplacements of individual languages, especially at the low level of language classification.

To examine whether the minor misplacement in low-level branches would affect our main results of linguistic relatedness of the five language branches, time depth, and dispersal center, we made the following settings during the reconstruction of language trees, respectively: (1) default settings (version in the manuscript); (2) constraining the languages of same groups as monophyletic groups, respectively (i.e., Kra, Hlai, Ong-Be, Kam-Sui, Southwestern Tai, Central Tai, and Northern Tai); (3) excluding Saek language from our data; (4) constraining the six varieties of Shan language as a monophyletic group. The first setting had no prior constrains and was the version used in this study. The second setting ensured Kra, Hlai, Ong-Be, Kam-Sui branches, and the three Tai groups to be monophyletic respectively and constraining Saek into Northern Tai group according to traditional linguists' views (Li, 1977, Gedney, 1989; Khanittanan and Yang 2003; Khanittanan and Yang 2004). The third setting excluded Saek because it was suggested as a borrowing-prone language that would undermine the overall shape of tree (Khanittanan and Yang 2003; Khanittanan and Yang 2004; Bouckaert, *et al.*, 2012; Greenhill and Gray, 2012). The fourth setting ensured the presence of Proto-Shan by constraining the Shan varieties (Brown, 1985; Zhou, 2001; He, 2003; Gogoi, *et al.*, 2020). Since Shan populations were influential in MSEA (He, 2003; He, 2007), the varieties of Shan language would experience substantial contacts with other languages and failed to form a monophyletic group. These four different settings would generate four sets of trees with

different low-level branching patterns, which would be then used for further analysis to compare linguistic relatedness, time-depth, and dispersal center. The model used for the reconstruction of trees was the combination of the Covarion model and the Relaxed Lognormal clock model, which was the best-fitted one under default settings. The studies for discrete phylogeographic inference were also consistent with that in the section “*Discrete phylogeographic inference*” in Methods. The results were illustrated in Figure R2, Figure R3, and Table R1. Accordingly, all results supported the linguistic relatedness proposed by Ostapirat (2017) with high posterior values; and were consistent in the time depth (around 4,000 years BP) as well as in other high-level internal nodes; and suggested that the coastal area was most likely to be the dispersal center of KD languages. Therefore, our replication with different combinations of model settings showed strong robustness of our main conclusions for KD languages.

Figure R2. Comparison of linguistic relatedness of the five language branches among versions of different settings. The maximum clade credibility trees under four versions of settings were

shown with the posterior values of every high-level node. Trees were reconstructed under the Covarion model and the Relaxed Lognormal clock model. (A) default settings (version in the manuscript); (B) constraining the languages of the same groups as monophyletic groups, respectively (i.e., Kra, Hlai, Ong-Be, Kam-Sui, Southwestern Tai, Central Tai, and Northern Tai); (C) excluding Saek from our data; (D) constraining the six varieties of Shan language as a monophyletic group.

Figure R3. Comparison of the root age distributions of KD languages among versions of different settings. Trees were reconstructed under the Covarion model and the Relaxed Lognormal clock model.

Table R1. Comparison of time depth and root probability among versions of different settings.

Version	(1)	(2)	(3)	(4)
Root age	4041	4124	4014	4030
Origination date of Hlai	3179	3224	3178	3162
Origination date of Ong-Be (Years BP)	2599	2615	2613	2584
Divergence of Kam-Tai	1950	1958	1935	1936
Divergence of Southwestern Tai	824	871	792	812
Coastal area	0.470	0.502	0.552	0.464
Guizhou	0.186	0.168	0.160	0.189
Root probability				
Yunnan	0.183	0.191	0.168	0.188
Hainan Island	0.065	0.064	0.059	0.055
MSEA	0.096	0.075	0.060	0.104

Second, Pereltsvaig and Lewis suggested that Bayesian Phylogenetic dating methods could be inaccurate due to misshapen trees and mistaken calibration points. Indeed, we could not exclude the possibilities that several internal nodes could be misplaced as indicated by the low posterior values. The low values related to one kind of phylogenetic uncertainty suggested that these internal nodes might be questionable, so the time estimation was meaningless for them. However, in our study,

most internal nodes were supported by high posterior values (>0.9), especially for the higher-level internal nodes. This indicated that our estimation for the relationship was robust, and thus, the time estimation was meaningful to infer the divergence time of proto-languages. Moreover, to make our calibration points more reasonable, we referred to previous ethnological and linguistic documents and used probability distributions instead of given years (see Table S2. Calibrations on Tree).

Third, we used a discrete phylogeographic approach to infer the dispersal center and pattern of KD languages, compared with that used by Gray-Atkinson's team. In Gray-Atkinson's method, the tree topology, time depth, and geographic distribution were estimated simultaneously. It might be defective because the long-distance language dispersal would produce long branches in the language phylogeny and further introduce considerable bias in the divergence time estimation given a questionable phylogenetic tree. However, in our study, we independently inferred the geographical distribution given the fixed tree topology. The fixed topology highlighted our discrete phylogeographic inference was independent of the divergence time estimation. Moreover, our approach was based on the ancestral discrete character reconstruction for the dispersal center inference, which has been widely used in biological evolution (Jönsson, *et al.*, 2011; Kehlmaier, *et al.*, 2023) and recently applied in language and cultural evolution (Currie *et al.*, 2010 and 2013; Opie *et al.*, 2014). With consideration of phylogenetic uncertainties, we did not use one phylogenetic tree topology in phylogeographic inference but used a series of trees sampled from the MCMC method. Even though using geographic coordinates of latitude and longitude would give a more precise location for ancestral nodes, we used administrative geographic regions instead. The defect of using latitude and longitude is that the spatial sampling pattern would largely affect the results. For example, the Beibu Gulf, northwest of the South China Sea, is surrounded by Mainland South China, Hainan Island, and MSEA. We could not exclude the possibility that the locations of several internal nodes are estimated in Beibu Gulf, which seems to be unreasonable. In contrast, using the region for estimation seems more objective and reasonable because the bias introduced by the spatial sampling pattern would be smoothed and the location of internal nodes would only be the given regions.

In summary, despite the flaws of applying Bayesian Phylogenetic methods to linguistics, these methods still remain reliable for quantitatively inferring language evolutionary history. In this study, we endeavored to relieve the influences caused by these flaws to ensure our results were tenable.

Regarding the concerns for our dataset, please see comments 7-9 below and section “*The summary of processing the lexical cognate identification*” in the revised version of Supplementary Information.

References

1. Pereltsvaig, A. & Lewis, M. W. *The Indo-European Controversy: Facts and Fallacies in Historical Linguistics*. (Cambridge University Press, 2015).
2. Bouckaert, R. *et al.* Mapping the origins and expansion of the Indo-European language family. *Science* **337**, 957-960 (2012).
3. Zhang, M., Zheng, H. X., Yan, S. & Jin, L. Reconciling the father tongue and mother tongue hypotheses in Indo-European populations. *Natl Sci Rev* **6**, 293-300 (2019).
4. Greenhill, S. J. *et al.* Evolutionary dynamics of language systems. *Proceedings of the National Academy of Sciences* **114**, E8822-E8829 (2017).
5. Thomason, S. G. & Kaufman, T. *Language contact*. (Edinburgh University Press, 2001).

- Atkinson, Q. D. & Gray, R. D.
6. Greenhill, S. J., Heggarty, P. & Gray, R. D. in *The handbook of historical linguistics* Vol. 2 *Bayesian Phylolinguistics* Ch. 11, 226-253 (2020).
 7. Ringe, D., Warnow, T. & Taylor, A. Indo-European and computational cladistics. *Transactions of the philological society* **100**, 59-129 (2002).
 8. Atkinson, Q. D. & Gray, R. D. Curious parallels and curious connections—phylogenetic thinking in biology and historical linguistics. *Systematic biology* **54**, 513-526 (2005).
 9. Greenhill, S. J. & Gray, R. D. Basic vocabulary and Bayesian phylolinguistics: Issues of understanding and representation. *Diachronica* **29**, 523-537 (2012).
 10. Bouckaert, R. R. & Heled, J. DensiTree 2: Seeing trees through the forest. *BioRxiv*, 012401 (2014).
 11. Li, F. K. *A handbook of comparative Tai*. (The University Press of Hawaii, 1977).
 12. Gedney, W. J. in *Michigan papers on South and Southeast Asia* Vol. 29 (Ann Arbor: Center for South and Southeast Asian Studies, 1989).
 13. Khanittanan, W. & Yang, G. Saek Language (Part One) (Chinese version). *Nankai Linguistics*, 154-181+187 (2003).
 14. Khanittanan, W. & Yang, G. Saek Language (Part Two) (Chinese version). *Nankai Linguistics*, 171-196 (2004).
 15. Brown, J. M. *From ancient Thai to modern dialects and other writings on historical Thai linguistics*. (White Lotus Company, 1985).
 16. Zhou, Y. *Daiyu Fangyan Yanjiu (Chinese version)*. (The Ethnic Publishing House, 2001).
 17. He, P. Miandian Fengjianwangchao Shili de Beikuo yu Danbang de Xingcheng (Chinese version). *South and Southeast Asian Studies*, 44-52 (2003).
 18. Gogoi, P., Morey, S. & Pittayaporn, P. The Tai Ahom sound system as reflected by the texts recorded in the bark manuscripts. *Journal of the Southeast Asian Linguistics Society* **13**, 14-42 (2020).
 19. He, P. The Changes of the Frontier Region of Southwest China and the Origin of the Shans in Burma (Chinese version). *Journal of Yunnan Minzu University (Philosophy and Social Sciences Edition)*, 84-89 (2007).
 20. Ostapirat, W. Kra-Dai in Southern China (Invited talk in the Nankai University). (Nankai University, 2017).
 21. Jønsson, K. A., Fabre, P.-H., Ricklefs, R. E. & Fjeldså, J. Major global radiation of corvid birds originated in the proto-Papuan archipelago. *Proceedings of the National Academy of Sciences* **108**, 2328-2333 (2011).
 22. Kehlmaier, C. *et al.* Ancient DNA elucidates the lost world of western Indian Ocean giant tortoises and reveals a new extinct species from Madagascar. *Sci Adv* **9**, eabq2574 (2023).
 23. Currie, T. E., Greenhill, S. J., Gray, R. D., Hasegawa, T. & Mace, R. Rise and fall of political complexity in island South-East Asia and the Pacific. *Nature* **467**, 801-804 (2010).
 24. Currie, T. E., Meade, A., Guillon, M. & Mace, R. Cultural phylogeography of the Bantu Languages of sub-Saharan Africa. *Proc Biol Sci* **280**, 20130695 (2013).
 25. Opie, C., Shultz, S., Atkinson, Q. D., Currie, T. & Mace, R. Phylogenetic reconstruction of Bantu kinship challenges Main Sequence Theory of human social evolution. *Proc Natl Acad Sci U S A* **111**, 17414-17419 (2014).

(7) - It is not clear how the data are coded. Table 1 contains many characters but no definition of the characters are given. So what are the "root-meanings" for the six characters with the concept "belly" for example.

Response: We apologize for the unclear description. We modified the structure of lexical data in the Excel file named "Table S1. Linguistic Data". In the revised version, we excluded the unnecessary columns and labels which were only helpful for our initial establishing of the lexical cognate data. And we added the descriptions of the columns in the "Note" sheet to make it easy to understand. A detailed work pipeline was shown in Comment 8 and added in line 52-105 in Supplementary Information.

(8) - The author says that they manually labeled the forms in the sampled language on the basis of their linguistic knowledge. It is not clear how this was done.

Response: Similar to the previous pipeline on the evolution of Sino-Tibetan languages (Zhang *et al.*, 2019), the work pipeline in this study could be summarized as the following three steps:

1. Firstly, we compiled and integrated several public databases of Starostin (<https://starling.rinet.ru/cgi-bin/main.cgi>), Peiros (1998), Gedney (1994, 1995, and 1997), Liang & Zhang (1996), some doctoral theses (e.g., Norquest, 2007; Chen, 2018), lots of available linguistic documents of KD languages and several first-hand linguistic field-works conducted by authors (see Table S1. Linguistic Data and Table S7. Data Resources). Then, we screened the entries according to the lexical items of the Swadesh 100-word list. Considering the lexical data integration and then historical linguistic comparison, IPA transcriptions of each word in KD languages were used.
2. Secondly, we compiled the list of regular sound correspondences from previous studies which have been used to reconstruct the Proto-languages of each subgroup (e.g., Proto-Kra: Ostapirat (1999); Proto-Kam-Sui including proto-Kam and proto-Sui: Thurgood (1988), Edmondson and Yang (1998), Zeng (1994), Ferlus (1996), Peiros (1998), Huang (2002), Ostapirat, (2006), Long (2018); Proto-Ong-Be: Chen (2018); Proto-Hlai: Matisoff (1988), Ostapirat, (2004), Norquest (2007); Proto-Tai: Li (1977), Ferlus (1990), Pittayaporn (2008 and 2009), Ostapirat (2013)). We also referred to the reconstruction of Proto-Kra-Dai by Liang and Zhang (1996) and Ilya Peiros (1998). In line with the requirements of the comparative method of historical linguistics, the linguistic evidence in these previous studies consisted not only of lexical etymologies and shared morphology but also several regular sound correspondences. Such evidence further helps us to identify and label the lexical cognates across all KD languages sampled in our study. Accordingly, the regular sound correspondences of KD languages involved four aspects: consonant, vowel, coda, and tone. Based on historical-comparative methods, we could compare the lexical items (especially morphemes) to find out whether a corresponding group presents in given languages and identify the cognates of each item in the Swadesh 100-word list. We separated the identified lexical cognates for each meaning item into different lines where each line referred to a distinct cognate set. The IPA transcriptions of lexical items of KD languages were listed in the corresponding lines of the cognate set when these languages shared the same cognate set (Table S1. Linguistic Data, sheet name "Lexical items"). In this sheet, we left the cell blank if the lexical cognate was absent and labeled 28 cells with "provisional". Finally, we obtained a raw data table including 100 languages (columns) and 646 cognate sets (rows).
3. Thirdly, the lexical cognate table was transformed into a binary-coding table (see Table S1.

Linguistic Data, sheet name “Binary coded set”). If a cognate set existed in a language, we labeled the cognate set in this language as “1”. If a cognate was not recorded in a language or was identified as a borrowing word, we considered that the cognate did not exist in this language and we would label a “0”. Specifically, when we could not confidentially identify a cognate set and consider it as a provisional one for a language, we would label the cognate as “?”, which means it would take “0” and “1” with equal probabilities in the computational procedure.

Here, we took the lexical item “eye” as an example. According to our knowledge of regular sound correspondences of “eye”, the consonants in Tai (t, pr, p(j), th, r(ɣ), etc.) (Li, 1977), Ong-Be (d) (Chen, 2018), Kam-Sui (t, ɲ, nd, l, etc.) (Edmondson and Yang, 1988; Ferlus, 1996), Hlai (ts^h, h, t) (Matisoff, 1988), and Kra (t, d, ð, ɕ) (Ostapirat, 1999) formed a diverse regular sound corresponding group. Meanwhile, the vowels and tones were neatly corresponding with each other, respectively. The vowels included a, ɛ, ɔ, their corresponding forms of vowel raising and diphthongization (e.g., au, əu, iu in Kra branch (Ostapirat, 1999)); whereas the tones were of tone 1 (A1), and the corresponding forms of tone divergence and merger (e.g., the change from A1 to A1', A2, B1, C2 in Tai branch (Li, 1977)). These correspondences indicated that they could share the same cognate set. However, only TnLianshan (Lianshan Zhuang dialect) was found to have the phonetic form “phat.8”. In contrast to the absence of plosive coda in other languages, “phat.8” had a plosive coda and was of tone 8 (D2). These differences showed that “phat.8” should be regarded as another cognate set. Therefore, after transforming to binary coded sets, there were two cognate sets of “eye”. The first cognate set was “1” for all languages except “0” for TnLianshan, and the second was “1” for TnLianshan but “0” for other languages.

References

1. Zhang, M., Yan, S., Pan, W. & Jin, L. Phylogenetic evidence for Sino-Tibetan origin in northern China in the Late Neolithic. *Nature* **569**, 112-115 (2019).
2. Peiros, I. *Comparative Linguistics in Southeast Asia*. (Pacific Linguistics, Research School of Pacific and Asian Studies, 1998).
3. Gedney, W. J. *William J. Gedney's Southwestern Tai Dialects: Glossaries, Texts and Translations (No. 42)*. (University of Michigan Press, 1994).
4. Gedney, W. J. *William J. Gedney's Central Tai Dialects: Glossaries, Texts, and Translations (No. 43)*. (University of Michigan Press, 1995).
5. Gedney, W. J. *William J. Gedney's Tai Dialect Studies: Glossaries, Texts, and Translations (No. 45)*. (University of Michigan Press, 1997).
6. Liang, M. & Zhang, J. R. *An Introduction to the Kam-Tai Languages (Chinese version)*. (China Social Sciences Press, 1996).
7. Norquest, P. K. *A phonological reconstruction of Proto-Hlai*. (The University of Arizona, 2007).
8. Chen, Y.-l. *Proto-Ong-Be*, University of Hawai'i at Manoa, (2018).
9. Ostapirat, W. *Proto-Kra*. (University of California, Berkeley, 1999).
10. Thurgood, G. Notes on the reconstruction of Proto-Kam-Sui. *Comparative Kadai: linguistic studies beyond Tai*, 179-218 (1988).
11. Edmondson, J. A. & Yang, Q. Word - initial Preconsonants and the History of Kam-Sui Resonant Initials and Tones. *Comparative Kadai: Linguistic studies beyond Tai*, 143-166 (1988).
12. Zeng, X. Y. *Hanyu Shuiyu Guanxici Yanjiu (Chinese version)*. (Chongqing Publishing Group,

- 1996).
13. Ferlus, M. Remarques sur le consonantisme du proto kam-sui. *Cahiers de linguistique-Asie Orientale* **25**, 235-278 (1996).
 14. Huang, Y. *Hanyu Dongyu Guanxici Yanjiu (Chinese version)*. (Tianjin Guji Press, 2002).
 15. Ostapirat, W. Alternation of tonal series and the reconstruction of Proto-Kam-Sui. *Dah-an Ho, H. Samuel Cheung, Wuyun Pan, & Fuxiang Wu(eds.), Linguistic studies in Chinese and Neighboring languages: Festschrift in Honor of Professor Pang-Hsin Ting on His 70th Birthday*, 1077-1121 (2006).
 16. Long, R.-T. Research on the Tone Merger of Kam Language (Chinese version). *Guizhou Ethnic Studies* **39**, 194-199 (2018).
 17. Matisoff, J. A. Proto-Hlai initials and tones: a first approximation. *Comparative Kadai: linguistic studies beyond Tai*, 289-321 (1988).
 18. Ostapirat, W. Proto-Hlai sound system and lexicons. *Studies on Sino-Tibetan languages: Papers in honor of Professor Hwang-cherng Gong on his seventieth birthday*, 121-175 (2004).
 19. Li, F. K. *A handbook of comparative Tai*. (The University Press of Hawaii, 1977).
 20. Ferlus, M. REMARQUES SUR LE CONSONANTISME DU PROTO THAI-YAY (Révision du proto tai de LI Fangkuei). *23rd International Conference on Sino-Tibetan Languages and Linguistics* (1990).
 21. Pittayaporn, P. Proto-Southwestern-Tai revised: A new reconstruction. *JSEALS*, 119 (2009).
 22. Pittayaporn, P. *The phonology of proto-Tai*, Cornell University, (2009).
 23. Ostapirat, W. The rime system of Proto-Tai. *Bulletin of Chinese Linguistics* **7**, 189-227 (2013).

(9) - Why are loanwords removed. They are not different from any other kinds of lexical innovations. Response: The loanwords are recognized to be confounding factors for any type of linguistic comparative method to reconstruct the genealogical classification of a language tree and need to be identified and removed before applying the historical-comparative method (Haspelmath, 2008; Pereltsvaig & Lewis, 2015).

In our lexical data of KD languages, the loanwords were classified into two groups: one did not comply with regular sound correspondences; the others complied with regular sound correspondences in specific branches or clades of KD languages. In particular, the former group should be excluded as it was clear to identify the source. For example, “mo:t5” of lexical item “One” of TswNgheAn could be a loanword from Vietnamese; and “nak3” of lexical item “Black” of TswAiton could be a loanword from Burmese.

In contrast, the latter group was much more complicated. It is generally accepted that KD languages experience a strong horizontal influence on Southern Chinese dialects. Even, in the Swadesh 100-word list, we could find that Chinese loanwords became replacing the indigenous words in some KD languages (Zhang, 1982, 1987 and 1988; Shao, 2016; Li & Wu, 2017; Wang *et al.*, 2020). For example, there existed numeral systems of Tai and Kam-Sui branches borrowed from that of Middle Chinese. Here, we took an example of the lexical item “One” in the Hlai, Ong-Be, Tai, and Kam-Sui branches. In particular, Hlai languages borrowed the phonetic form of “One” from the Hainan Min dialect; the Jizhao dialect of Ong-Be languages borrowed from the Wuchuan Yue dialect (Shao, 2016; Li & Wu, 2017; Wang *et al.*, 2020); Tai languages borrowed from Guangxi Pinghua dialect (Zhang, 1982, 1987 and 1988); and Kam-Sui languages borrowed from either Guangxi Pinghua Dialects or Southwestern Mandarin (Zhang, 1988). Although such systematic

lexical borrowings between KD languages and Chinese could be traced back to Middle Chinese, these were still excluded in our study due to the heterogeneous sources. In addition, the case of “Heart” in the Kam-Sui branch, Central Tai and Northern Tai clades, and the case of “White” in the Central Tai clade was also regarded as loanwords from Chinese dialects. Notably, Zhang (1982, 1987, and 1988) pointed out that these loanwords were mainly derived from the Guangxi Pinghua dialect in the late stage of KD language formation. Despite the presence of systematical regular sound correspondences in several branches and clades, we still excluded these loanwords in practice. Notably, we should not exclude some words which could be borrowed from Old Chinese, because they could form regular sound correspondences during the evolutionary process of KD languages. Therefore, these potential loanwords could be assimilated following the intrinsic sound changes within KD languages.

All in all, we excluded the loanwords borrowed from Middle Chinese and Modern Chinese dialects and maintained the potential loanwords borrowed from Old Chinese.

References

1. Haspelmath, M. Loanword typology: Steps toward a systematic crosslinguistic study of lexical borrowability. *Empirical Approaches to Language Typology*, 43-62 (2008).
2. Pereltsvaig, A. & Lewis, M. W. *The Indo-European Controversy: Facts and Fallacies in Historical Linguistics*. (Cambridge University Press, 2015).
3. Zhang, J. R. Guangxi Zhongnanbu Diqu Zhuangyu zhong de Lao Jieci Yuanyu Hanyu Gu "Pinghua" Kao. *Studies in Language and Linguistics*, 197-219 (1982).
4. Zhang, J. R. Guangxi Pinghua zhong de Zhuangyu Jieci (Chinese version). *Studies in Language and Linguistics*, 185-189 (1987).
5. Zhang, J. R. Guangxi Pinghua dui Dangdi Zhuangdong Yuzu Yuyan de Yingxiang (Chinese version). *Minority Languages of China*, 51-56 (1988).
6. Shao, L. *A Study of Jizhao Language in Guangdong (Chinese version)*, GuangXi University for Nationalities, (2016).
7. Li, J. F. & Wu, Y. A Grammatical Sketch of the Jizhao Language Spoken in Wuchuan of Guangdong Province (Chinese version). *Minority Languages of China*, 77-96 (2017).
8. Wang, W., Fu, C. & Wei, Y. Revisiting the Family of Jizhaohai Dialect (Chinese version). *Bulletin of Linguistic Studies*, 391-404+447 (2020).

(10) - Is the tree rooted? The author didn't seem to have included as outgroup. Why not? How would that affect the analysis?

Response: The phylogenetic tree of KD languages was a rooted time tree in our study inferred BEAST program. BEAST aims specifically at inferring rooted time trees, and uncertainty of time estimates, which sets it apart from other Bayesian packages that target unrooted trees. According to the book “Bayesian Evolutionary Analysis with BEAST” (Drummond & Bouckaert, 2015), the original test for methodological descriptions on Pages 102-103 showed that “*However adding an outgroup is generally discouraged in Bayesian time tree analyses because the inclusion of outgroups can introduce long branches which can make many estimation tasks more difficult...*”. In the practice of the BEAST program, “*...A Bayesian time-tree analysis will sample the root position along with the rest of the tree topology. If you then calculate the proportion of sampled trees that have a particular root, you obtain a posterior probability for the root position....*” Accordingly, we were

not allowed to include any outgroup in the Bayesian phylogenetic reconstruction of the time tree when using BEAST program, and the time tree was rooted.

Reference

1. Drummond, A. J. & Bouckaert, R. R. *Bayesian Evolutionary Analysis with BEAST*. (Cambridge University Press, 2015).

(11) Is there enough detail provided in the methods for the work to be reproduced?

No. See the methodological problems raised above.

Response: We summarized our workflow of identifying cognates in Comment 8 and performed robustness analyses of our results using Bayesian phylogenetic and phylogeographic methods (see Comment 6).

In addition, we provided the nexus file for most phylogenetic programs such as *SplitsTree* program and XML file for *BEAST* program aiming for replicating our results (Supplementary Data 1). Moreover, we also provided the input files for *BayesTraits* program (Supplementary Data 2) for replicating the phylogeographic inference of the dispersal center; and MATLAB codes (Supplementary Data 3) for reproducing the Figure 3 in the manuscript. These files were uploaded as supplementary files along with the manuscript submission.

Reviewer 2: I have also included two files with specific comments. I hope you find them useful. Specific comments on the Main Text (attachment 1):

(12) Line 143: comment “What is this” on “root-meaning”.

Response: We apologize for the unclear description. In the revised Main Text and Supplementary Information, we replaced “root-meaning” with the word “cognates” or “lexical cognates”, which will be less ambiguous for readers with expertise in linguistics.

(13) Line 159: comment “Is this a lecture? Any publication?” on “Reference 35”.

Response: It was an invited talk by Weera Ostapirat at Nankai University in 2017. We added more reference information for this reference.

(14) Line 159-165: comment “Time estimation is a difficult issue in linguistic reconstruction. Any argument for why methods used in genetic is applicable to language? Does rate of diversification works the same way as rate of genetic diversification?” on “The estimated divergence time indicated that the first split of KD languages occurred around 4,000 years ago (median value = 4,084 years BP), and its 95% HPD interval range was from approximately 2,700 to 5,500 years ago (Figure 1C). The estimated time was significantly lower than Liang, Zhang, and Li’s expectation of KD divergence over 5,000 years ago ($t = -114.16$, $p\text{-value} < 2.2e-16$, Figure S5), but is majorly compatible with Ostapirat and Peiros’s considerations. The initial divergence time estimations of KD languages under different model combinations were compatible with each other (Figure S6).”.

Response: In our study, we applied the phylogenetic methods to the lexical cognate data which is originally used in evolutionary biology to infer the diversification of species. Phylogenetic analyses model explicitly the process of linguistic descent by which any family of languages diverges out of its common ancestor, similar to the divergence of species from their ancestors in biology. This is why they are also applicable to language in principle (Greenhill, *et al.*, 2020). Despite the parallel

evolution and several shared conceptual similarities between the biological system and language system, genes and languages could exhibit different evolutionary processes in reality. However, it was noted that the recent advances in phylogenetic methods were not designed solely for inferring genetic evolution. Apart from the traditional function, various models have been developed and other traits can be used in the phylogenetic analysis. The input data include morphological characters (Nylander, *et al.*, 2004; Glenner, *et al.*, 2004; Ni, *et al.*, 2021), cultural traits (O'Brien, *et al.*, 2001; Tehrani, *et al.*, 2002; Matthews, *et al.*, 2011), and linguistic data (often lexical cognate) (Bouckaert *et al.*, 2018; Kolipakam *et al.*, 2018; Sagart *et al.*, 2019). For the divergence time estimation, these methods calibrate the clock in both genetic and linguistic trees in the same manner. They use fossil evidence in biology (historical information in linguistics) to specify the age of known clades (Nascimento, *et al.*, 2017; Greenhill, *et al.*, 2020). Then, these calibrations allow the analysis to estimate the clock rates in regions where the timing is known, and to extrapolate these to infer rates in regions where the timing is unknown (Greenhill, *et al.*, 2020).

References:

1. Greenhill, S. J., Heggarty, P. & Gray, R. D. in *The handbook of historical linguistics* Vol. 2 *Bayesian Phylolinguistics* Ch. 11, 226-253 (2020).
2. Nylander, J. A. A., Ronquist, F., Huelsenbeck, J. P. & Nieves-Aldrey, J. Bayesian Phylogenetic Analysis of Combined Data. *Systematic Biology* **53**, 47-67 (2004).
3. Glenner, H. et al. Bayesian inference of the metazoan phylogeny; a combined molecular and morphological approach. *Curr Biol* **14**, 1644-1649 (2004).
4. Ni, X. *et al.* Massive cranium from Harbin in northeastern China establishes a new Middle Pleistocene human lineage. *Innovation (Camb)* **2**, 100130 (2021).
5. O'Brien, M. J., Darwent, J. & Lyman, R. L. Cladistics Is Useful for Reconstructing Archaeological Phylogenies: Palaeoindian Points from the Southeastern United States. *Journal of Archaeological Science* **28**, 1115-1136 (2001).
6. Tehrani, J. & Collard, M. Investigating cultural evolution through biological phylogenetic analyses of Turkmen textiles. *Journal of Anthropological Archaeology* **21**, 443-463 (2002).
7. Matthews, L. J., Tehrani, J. J., Jordan, F. M., Collard, M. & Nunn, C. L. Testing for divergent transmission histories among cultural characters: a study using Bayesian phylogenetic methods and Iranian tribal textile data. *PLoS One* **6**, e14810 (2011).
8. Bouckaert, R. R., Bowern, C. & Atkinson, Q. D. The origin and expansion of Pama-Nyungan languages across Australia. *Nature ecology & evolution* **2**, 741-749 (2018).
9. Kolipakam, V. *et al.* A Bayesian phylogenetic study of the Dravidian language family. *Royal Society open science* **5**, 171504 (2018).
10. Sagart, L. *et al.* Dated language phylogenies shed light on the ancestry of Sino-Tibetan. *Proceedings of the National Academy of Sciences of the United States of America* **116** (2019).
11. Nascimento, F. F., Reis, M. D. & Yang, Z. A biologist's guide to Bayesian phylogenetic analysis. *Nat Ecol Evol* **1**, 1446-1454 (2017).

(15) Line 168-171: comment “Why?” on “These areas were the Guangdong-Guangxi area named the coastal area, two separated inland areas of Yunnan and Guizhou provinces, the island area of Hainan province, and the MSEA covering other areas including Thailand, Vietnam, Laos, Myanmar, and India in this study, respectively.”.

Response: In our study, the phylogeographic analyses concentrated on examining two hypotheses of KD language dispersal. Therefore, the geographic area in South China and MSEA were divided into five distinct areas following the criteria of boundaries of administrative districts. We merged Guangxi Zhuang Autonomous Region and Guangdong Province as one area because only three languages (the Jizhao dialect of Ong-Be language, Lianshan Zhuang language, and Biao language) were sampled in Guangdong Province. As South China is regarded as the dispersal center of KD languages in previous studies (Gong, 2002; Kutanan, *et al.*, 2018; Wang, *et al.*, 2021), we thus merged the countries in MSEA as a single area.

References:

1. Gong, Q. H. *A Study of Chronological Strata of Sino-Thai Corresponding Lexical Items (Chinese version)* 200 (Fudan University Press, 2002).
2. Kutanan, W. *et al.* New insights from Thailand into the maternal genetic history of Mainland Southeast Asia. *European Journal of Human Genetics* **26**, 898-911 (2018).
3. Wang, M. G. *et al.* Reconstructing the genetic admixture history of Tai-Kadai and Sinitic people: Insights from genome-wide SNP data from South China. *Journal of Systematics and Evolution* (2021).

(16) Line 173: comment “To me, this looks pretty low. What is the benchmark?” on “a maximum probability of 50.6%”.

Response: The practice of ancestral discrete character reconstruction for the origin or dispersal center inference has been widely used in evolutionary biology (Jönsson, *et al.*, 2011; Kehlmaier, *et al.*, 2023) and recently applied in language and cultural evolution (Currie *et al.*, 2010 and 2013; Opie *et al.*, 2014). Using the phylogeographic approach, we reconstructed the ancestral character state at the root of given the KD language tree. We estimated the probabilities of five distinct areas with considerations of phylogenetic uncertainties. For the null hypothesis of the phylogeographic model, we could observe 20% for each distinct area which was inferred as a dispersal center equiprobably. Here, our results showed that the Guangdong-Guangxi area should be a dispersal area with a maximum probability of 47.0%, which was significantly higher than 20% in the revised version.

In addition, the MCMC method would give us a set of probabilities for each area, which could be tested by the Wilcoxon signed rank test to find whether there are significant differences among the sets of probabilities of the five distinct areas. We also used a bar plot to show the values and standard deviation of probabilities. Accordingly, we suggested that coastal area should be the dispersal center of KD languages with a significantly higher probability than other areas.

Figure R4. Probabilities of geographical distribution for the root of KD languages. Bars represented the standard deviation. The significance tested by Wilcoxon signed rank test was indicated by symbol “*”. “***” represented $p < 0.001$.

References:

1. Jønsson, K. A., Fabre, P.-H., Ricklefs, R. E. & Fjeldså, J. Major global radiation of corvid birds originated in the proto-Papuan archipelago. *Proceedings of the National Academy of Sciences* **108**, 2328-2333 (2011).
2. Kehlmaier, C. *et al.* Ancient DNA elucidates the lost world of western Indian Ocean giant tortoises and reveals a new extinct species from Madagascar. *Sci Adv* **9**, eabq2574 (2023).
3. Currie, T. E., Greenhill, S. J., Gray, R. D., Hasegawa, T. & Mace, R. Rise and fall of political complexity in island South-East Asia and the Pacific. *Nature* **467**, 801-804 (2010).
4. Currie, T. E., Meade, A., Guillon, M. & Mace, R. Cultural phylogeography of the Bantu Languages of sub-Saharan Africa. *Proc Biol Sci* **280**, 20130695 (2013).
5. Opie, C., Shultz, S., Atkinson, Q. D., Currie, T. & Mace, R. Phylogenetic reconstruction of Bantu kinship challenges Main Sequence Theory of human social evolution. *Proc Natl Acad Sci U S A* **111**, 17414-17419 (2014).

(17) Line 175: comment “Is this published?” on “Reference 21”.

Response: The reference was originally a doctoral thesis of Qunhu Gong in 2001, and has been published as a book (ISBN: ISBN7-309-03427-9/H.659) in Fudan University Press in 2002.

(18) Line 184: comment “Your citations only include genetic studies. What archeological evidence is there? We cannot tell whether the people of an archeological site were KD-speaking” on “archaeological evidence”.

Response: We apologized for the ambiguous description. We agree with the reviewer’s comments

that it is difficult to determine whether the people of the archeological site were KD-speaking populations based on genetic relatedness. As there is no obvious archaeological correlate for KD-speaking populations, we could only infer whether the archaeological sites belonged to KD people based on the shared genetic ancestry between ancient people and present-day KD people. Previous citations focused on genetic evidence and linked the ancient people with farming culture, but no direct evidence for archeological sites. We revised the original word “archeological” to “cultural”, and added two more references as followed. These references provided another cultural evidence (i.e., hanging coffins), indicating a demographic scenario consistent with that inferred from linguistic and genetic evidence:

1. Zhang, X. *et al.* A Matrilineal Genetic Perspective of Hanging Coffin Custom in Southern China and Northern Thailand. *iScience* **23**, 101032 (2020).
2. Wang, T. *et al.* Human population history at the crossroads of East and Southeast Asia since 11,000 years ago. *Cell* **184**, 3829-3841. e3821 (2021).

(19) Line 192-193: comment “Vague” on “the initial KD language divergence might be associated with the demographic activities of KD-speaking peoples”.

Response: We added more related context in the revised manuscript to make it clear. Please see line 267-272 in the revised version of the main text: “*During the 4.2K event, in the deteriorating environment, KD-speaking populations of maternal lineages grew slowly and the number of archaeological sites changed dramatically in Southern China regions. These implied that KD-speaking populations could experience the migration process due to climate change. These findings suggested that the initial divergence of KD languages might be coupled with the climate-induced demographic activities (e.g., migration) of KD-speaking populations.*”

(20) Line 244-245: comment “How? We do not have a reliable Proto-KD reconstruction” on “identified the genuine root-meaning traits which were inherited from a common ancestor of KD languages without lexical borrowings”.

Response: In practice, we referred to the reconstructed protoforms of etyma (or cognates) for the proto-KD language and its five branches from different bibliographies as follows:

Proto-Kra-Dai: Liang and Zhang (1996), Peiros (1998);

Proto-Kra: Ostapirat (1999);

Proto-Kam-Sui (including proto-Kam and proto-Sui): Thurgood (1988), Edmondson and Yang (1998), Zeng (1994), Ferlus (1996), Peiros (1998), Huang (2002), Ostapirat, (2006), Long (2018);

Proto-Ong-Be: Chen (2018);

Proto-Hlai: Matisoff (1988), Ostapirat, (2004), Norquest (2007);

Proto-Tai: Li (1977), Ferlus (1990), Pittayaporn (2008 and 2009), Ostapirat (2013)).

These bibliographies not only provide the protoforms but also enable us to summarize regular sound correspondences.

References:

1. Liang, M. & Zhang, J. R. *An Introduction to the Kam-Tai Languages (Chinese version)*. (China Social Sciences Press, 1996).
2. Peiros, I. *Comparative Linguistics in Southeast Asia*. (Pacific Linguistics, Research School of Pacific and Asian Studies, 1998).

3. Ostapirat, W. *Proto-Kra*. (University of California, Berkeley, 1999).
4. Thurgood, G. Notes on the reconstruction of Proto-Kam-Sui. *Comparative Kadai: linguistic studies beyond Tai*, 179-218 (1988).
5. Edmondson, J. A. & Yang, Q. Word - initial Preconsonants and the History of Kam-Sui Resonant Initials and Tones. *Comparative Kadai: Linguistic studies beyond Tai*, 143-166 (1988).
6. Zeng, X. Y. *Hanyu Shuiyu Guanxici Yanjiu (Chinese version)*. (Chongqing Publishing Group, 1996).
7. Ferlus, M. Remarques sur le consonantisme du proto kam-sui. *Cahiers de linguistique-Asie Orientale* **25**, 235-278 (1996).
8. Huang, Y. *Hanyu Dongyu Guanxici Yanjiu (Chinese version)*. (Tianjin Guji Press, 2002).
9. Ostapirat, W. Alternation of tonal series and the reconstruction of Proto-Kam-Sui. *Dah-an Ho, H. Samuel Cheung, Wuyun Pan, & Fuxiang Wu(eds.), Linguistic studies in Chinese and Neighboring languages: Festschrift in Honor of Professor Pang-Hsin Ting on His 70th Birthday*, 1077-1121 (2006).
10. Long, R.-T. Research on the Tone Merger of Kam Language (Chinese version). *Guizhou Ethnic Studies* **39**, 194-199 (2018).
11. Chen, Y.-l. *Proto-Ong-Be*, University of Hawai'i at Manoa, (2018).
12. Matisoff, J. A. Proto-Hlai initials and tones: a first approximation. *Comparative Kadai: linguistic studies beyond Tai*, 289-321 (1988).
13. Ostapirat, W. Proto-Hlai sound system and lexicons. *Studies on Sino-Tibetan languages: Papers in honor of Professor Hwang-cherng Gong on his seventieth birthday*, 121-175 (2004).
14. Norquest, P. K. *A phonological reconstruction of Proto-Hlai*. (The University of Arizona, 2007).
15. Li, F. K. *A handbook of comparative Tai*. (The University Press of Hawaii, 1977).
16. Ferlus, M. REMARQUES SUR LE CONSONANTISME DU PROTO THAI-YAY (Révision du proto tai de LI Fangkuei). *23rd International Conference on Sino-Tibetan Languages and Linguistics* (1990).
17. Pittayaporn, P. Proto-Southwestern-Tai revised: A new reconstruction. *JSEALS*, 119 (2009).
18. Pittayaporn, P. *The phonology of proto-Tai*, Cornell University, (2009).
19. Ostapirat, W. The rime system of Proto-Tai. *Bulletin of Chinese Linguistics* **7**, 189-227 (2013).

(21) Line 247: comment “No way to trace what each lexical character was. The authors should include actual reconstructed forms so that 1) readers know which “root-meaning” item the character represents and 2) judge whether the root-meaning coding is accurate. This is especially important since there is no generally accepted reconstructed lexicon of Proto-KD.” on “653 binary-coded root-meanings”.

Response: Please see our workflow in Comment 8 and the table sheet of table description (named “Note”) in Table S1. Linguistic data.

(22) Line 282: comment “Does this introduce a bias? Why not do the same with other languages? Is the evidence strong?” on “According to the available historical ethnic records”.

Response: This can be regarded as a correction and no bias would be introduced. We did not do the same with other languages because the default setting is to add no constraints on ancestral states. However, we added a fossil state constraint for the most recent common ancestor of Ong-Be

languages because there existed a geographical sampling unbalance where only one language sample (Jizhao dialect) persisted in the coastal area whereas other Ong-Be languages are distributed on Hainan Island. Such unbalance in the geographical distribution of Ong-Be would largely affect the estimation of the ancestral state. Our setting conformed with the ethnological and linguistic views. According to the master's thesis of Zhang (Zhang, 2016), the ancestors of the Ong-Be languages arrived on Hainan Island through the Leizhou Peninsula from western Guangdong. Other Chinese ethnolinguists were also in favor of this view such as Min Liang and Junru Zhang (Zhang and Liang, 1996).

References

1. Zhang, J. R. A Study on the Relationship between Wuchuan people in Western Guangdong and Lingao people in Hainan from the Cultural Perspective of Regional Names (*Chinese version*), Guangdong Polytechnic Normal University (2016).
2. Liang, M. & Zhang, J. R. An Introduction to the Kam-Tai Languages (*Chinese version*). 1-49 (China Social Science Press, 1996).

Comments on the Supplementary Information (attachment 2):

(23) Line 50-52: comment “For most linguists, this is called “cognates.” If you want to use the concept of “root-meaning”, make sure you explain it more clearly.” on “roots or etyma inherited in direct descent from an etymological ancestor in a common parent language (i.e., a proto-language).”

Response: In the revised manuscript, we replace “root-meanings” with “cognates” or “lexical cognates” to avoid ambiguity.

(24) Line 63-65: comment “How?” on “We used the traditional comparative methods and internal reconstruction in historical linguistics to annotate the lexical etyma and identify the root meanings across KD languages according to the proto-forms.”

Response: The workflow was summarized in Comment 8.

(25) Line 65: comment “Why is this?” on “stratum analysis”.

Response: In historical linguistic comparison, one important task is to distinguish horizontal transmission from vertical transmission. In our study, the approach of historical strata analysis (Ding, 2007) was performed to identify the lexical borrowings of KD languages from Middle Chinese (e.g., Zhuang language) and modern Chinese dialects (e.g., Jizhao dialect of Ong-Be language). In the revised manuscript, we replaced “stratum analysis” with “historical strata analysis in linguistics”, and modified the corresponding sentence expression to improve the readability of the manuscript.

Reference

1. Ding, B. X. *Chinese Dialects and Historical Strata (Chinese version)*. (Shanghai Educational Publishing House, 2007).

(26) Line 73-74: comment “So they are excluded?” on “must distinguish each morpheme of every lexical item as a single etymon. For example, the lexical item of “Sun” can be represented by a word that is composited of two etyma "eye" and "day"”.

Response: No, we did not exclude such kind of lexical item. For each multi-morpheme compound word given in a language, firstly, we divided the word into distinctive morphemes and identify their etyma. Then, we found out which etymon presented most in all KD languages. If this etymon is not in the Swadesh-100 core vocabulary list, it would substitute the original lexical item to be applied to comparative methods. For example, “sun” is a single morpheme in Indo-European languages, but a multi-morpheme compound word consisted of the etymon “day” in most KD languages. As “day” is not in the Swadesh-100 core vocabulary list, we replaced “sun” with “day” in most languages. For cognate identification, for example, we first obtain the Proto-Tai root *ɲwan for the etymon “day” from the previous linguistic studies. In PuBiao, we observe the multi-morpheme compound word qa33 la:ŋ53 in our study of the lexical item of “sun”. In this word, la:ŋ53 could be the core morpheme related to “sun” because qa33 is probably the nominalization of “light”, which is inconsistent with the etymon “sun”. Thus, we can linguistically compare the PuBiao’s form la:ŋ53 with the Proto-Tai root *ɲwan, and then conclude that they are not the same cognate set.

(27) Line 86-89: comment “Citations?” on “The third one is that Chinese loanwords have been excluded from the KD database. It has well known that Kra-Dai languages receive strong influence from Chinese, especially Middle Chinese and modern southern Chinese dialects (e.g., Cantonese and Pinghua). Even, Chinese loanwords can also be found in the basic vocabulary of KD languages such as "one", "white" and "heart" etc”.

Response: We added the following citations in the revised version of Supplementary Information.

1. Zhang, J. R. Guangxi Zhongnanbu Diqu Zhuangyu zhong de Lao Jieci Yuanyu Hanyu Gu "Pinghua" Kao (Chinese version). *Studies in Language and Linguistics*, 197-219 (1982).
2. Zhang, J. R. Guangxi Pinghua zhong de Zhuangyu Jieci (Chinese version). *Studies in Language and Linguistics*, 185-189 (1987).
3. Zhang, J. R. Guangxi Pinghua dui Dangdi Zhuangdong Yuzu Yuyan de Yingxiang (Chinese version). *Minority Languages of China*, 51-56 (1988).
4. Shao, L. *A Study of Jizhao Language in Guangdong (Chinese version)*, GuangXi University for Nationalities, (2016).
5. Li, J. F. & Wu, Y. A Grammatical Sketch of the Jizhao Language Spoken in Wuchuan of Guangdong Province (Chinese version). *Minority Languages of China*, 77-96 (2017).
6. Wang, W., Fu, C. & Wei, Y. Revisiting the Family of Jizhaohai Dialect (Chinese version). *Bulletin of Linguistic Studies*, 391-404+447 (2020).

(28) Line 96-99: comment “There’s no principled way to exclude only relatively old and relatively recent loanwords. And why would you exclude loans since borrowing is one mechanism that introduces lexical innovations? How about non-Sinitic borrowing?” on “Meanwhile, several lexica such as “head” and “work” are *allegedly* loanwords borrowed from Old Chinese. However, we do not exclude these lexicons because as the early loanwords, these lexica appear in most of the KD languages and form regular sound correspondence with other languages”

Response: We apologize for the unclear description. We modified the sentence into “Meanwhile, several lexica such as “head” and “work” are allegedly loanwords borrowed from Old Chinese. In our study, we maintained these loanwords in the lexical data because these lexica appear in most of the KD languages and form sound correspondence with other languages.” The criteria for identifying borrowing are described in Comment 9.

It is acknowledged that the source of borrowings is not limited to Sinitic languages. KD languages could borrow from other regionally dominant languages distributed in South China and MSEA. These languages include Austroasiatic languages (e.g., Khmer, Vietnamese), Tibetan-Burman languages (e.g., Burmese, Yi language), Hmong-Mien languages (e.g., Hmong), and even other KD languages (e.g., Bouyei, Thai) (Ni, 1990). However, the studies on the influence of non-Sinitic languages on KD languages was lacking and was only at the initial stage. We have endeavored to exclude the well-identified borrowings, such as the examples in Comment 9.

Reference

1. Ni, D. *An introduction to Kam-Tai languages (Chinese version)*. (China Minzu University Press, 1990).

(29) Line 151-157: comment “How do you know they were KD speaking.” on “Hosner *et al.* (URL: <https://doi.pangaea.de/10.1594/PANGAEA.860072>). Hosner’s data contains a total of 51,074 archaeological sites from the early Neolithic to the early Iron Age (about 10,000 - 2,000 years BP) with a spatial extent covering most regions of China. The information of each site included the cultural name (e.g., Liangzhu and Tanshishan cultures), time range (max, min, average), and geographical location (province, longitude, and latitude). Their data were integrated from three major campaigns of systematic archaeological surveys waged by the Chinese government in 1956, 1981, and 2007”

Response: Considering the genetic (Wang *et al.*, 2021; Chen *et al.*, 2022) and paleogenetic (Wang *et al.*, 2021) evidence of the population continuity of the KD-speaking population in South China, we used the number of archaeological sites to show a trend of population growth for all ethnic groups in South China before the common era. The geographic distributions of archaeological sites used for interdisciplinary alignment covered the major places of residence of modern KD-speaking populations. From a global perspective, the dynamic change in the number of archaeological sites could be regarded as an approximate demographic dynamic of all populations in South China where KD-speaking people lived in. We acknowledged that the number of archaeological sites could only serve as a coarse indicator but was already an effective one to our knowledge.

References

1. Wang, M. G. *et al.* Reconstructing the genetic admixture history of Tai-Kadai and Sinitic people: Insights from genome-wide SNP data from South China. *Journal of Systematics and Evolution* (2021).
2. Chen, H. *et al.* Tracing Bai-Yue Ancestry in Aboriginal Li People on Hainan Island. *Mol Biol Evol* **39** (2022).
3. Wang, T. *et al.* Human population history at the crossroads of East and Southeast Asia since 11,000 years ago. *Cell* **184**, 3829-3841 e3821 (2021).

(30) Line 228: comment “Pittayaporn thinks CT is not monophyletic.” on “complied with Pittayaporn’s preliminary classification system”.

Response: Thank you for pointing it out. In previous studies, Edmondson and Pittayaporn proposed that CT could not be monophyletic. And our typological structure of the KD phylogenetic tree also showed the same conclusion as their claims. We used the new expression “*Moreover, Edmondson*

showed a much more diversified Central Tai phylogeny with a computational phylogenetic analysis, suggesting that CT is not monophyletic and is split up into multiple branches, which complied with Pittayaporn's preliminary classification." instead of the original sentence. See Line 283-286 in the revised Supplementary Information.

(31) Line 251-254: comment "This result shows that the methodology is problematic since it cannot detect lexical borrowing among KD languages." on "Noted that the Saek as a Northern Tai language was grouped into the Southwestern Tai languages. It suggested that Saek could experience substantial lexical borrowings or replacements in the basic vocabulary from its surrounding Southwestern Tai languages."

Response: The placement of Saek was further discussed in part (a) of Comment 5 and in Line 350-371 in the section "*Phylogenetic topology of the KD languages*" in the revised version of Supplementary Information.

REVIEWER COMMENTS

Reviewer #1 (Remarks to the Author):

I have no further comments

REVIEWER #2, comments:

I think the authors did an excellent job responding to my comments. I especially appreciate how they acknowledge theoretical and methodological issues that still need to be addressed, and how the results should be interpreted critically. I only have a couple of comments that I think the author should pay attention to.

- p.6 Status of Kra as a primary branch of Kra-Dai. The author explains that the observation that Kra is a primary branch is "consistent with the findings of previous historical linguistic works." The authors only rely on a small set of outdated and rather speculative/very preliminary studies. There is a lot more uncertainty and more diversity of opinion in the field of KD historical linguistics. The authors should check out Norquest's chapter and references therein.
<https://www.degruyter.com/document/doi/10.1515/9783110558142-013/html?lang=en>
- p.9 Assessing the results with existing linguistic knowledge. The author has done a good job. A very minor point is the use of "shared cognate sets" to mean set of forms inherited from a common ancestor or borrowing from other languages to be misleading. In historical linguistics "cognates" only means the former. To refer to both types of forms, something like "potential cognates" are preferable.
- p.13. The author assumes that Xing (1989) and Zhou and Luo (2001) are correct in classifying Yuanjiang as SWT. This is not the position linguists outside of Chinese take. Phonological innovations (as well as lexical innovations) in Yuanjiang show clearly that it is not part of SWT, despite it being "politically" classified as a Dai dialect. One of the many telling traits is the change /h-/ in reflex in the word for 'eye' (and other words with the same initial consonants in Proto-Tai) in contrast to /t-/ in SWT languages, cf. Li (1960)'s tentative classification of Tai languages among others.

Reviewer #3 (Remarks to the Author):

This is a very promising paper that provides an integrated perspective into the evolution and dispersal of the Kra-Dai language family. I enjoyed reading this and think it will become an important paper.

The authors are to be commended for making their data, analysis scripts, code and XML all available.

I have two major recommendations one regarding the use of the inaccurate HME estimator for model comparison, and one regarding the lack of detail about the integration genetic and archaeological analysis. The remainder of my recommendations are not as critical but would be nice to have.

I recommend this paper be accepted after minor revisions.

Major Recommendations:

1. Provide details on the model comparison, and fix if the HME in Tracer was used.

I could not find any details on how the phylogenetic model comparison was done, beyond the authors calculating Bayes Factors. How was the marginal likelihood estimated? Looking at the

supplementary table, it suggests that the authors used the Harmonic Mean Estimator implemented in Tracer. However the HME is known to provide inaccurate results as it is severely biased and substantially overestimated the true marginal likelihood (see Xie et al. 2011, *Sys. Biol*) which may mean that the model chosen as best is not the best. Current best practice is to use a formal model comparison method like Path Sampling (e.g. Beale et al. 2013, *Mol. Biol. Evol*) or Nested Sampling (e.g. Russel et al. 2018, *Sys. Biol*).

2. Integrate information about the mitochondrial and archaeological analyses into main text:

The mtDNA and archaeological analysis is interesting and really bolster the author's arguments about the underlying demographic dispersal. However, these are not mentioned in the main text and it was only when I read the methods that I realised these were there. Please integrate these findings into the main narrative as without them the final parts of the discussion read as far more speculative than they are (and I was going to flag this as one of my major complaints until I found these details in the methods/results).

As it stands, the current manuscript spends a LOT of time talking about one minor language, Saek. I appreciate the care given to evaluation of the Saek language but takes up a lot of space in main text. If space is an issue then I think that much of this Saek related content could be moved into the supplement (with an appropriate callout in main text), to enable the interdisciplinary results narrative to be highlighted (much more interesting than one language).

Minor recommendations:

1. Please provide a fuller discussion of the preferred homeland. The best-fitting homeland location only captures 47% of the probability mass, while two other homelands (Guizhou and Yunnan) have appreciable probabilities of ~20%. Given that these are also potentially likely candidates for homelands for this family, please provide some discussion of this and how -- if at all -- changing this homeland would affect the downstream interpretations in the paper.

2. The authors say they use the linguistic comparative method and there are some nice examples in the supplementary material of where they have done this. However, the state of Kra-Dai historical linguistics is not as well-established as many other families. Can the authors provide a table of their established sound correspondences to enable future work to extend and build off their work here?

Comments and Suggestions:

1. The authors abbreviation "KD" makes the paper more opaque, all to save 4 letters. It's unnecessary, please remove.

2. The link to 4.2k event is interesting, but the Effective Population size from the mitochondrial genomes appears to show no effect of the 4.2k event, unlike the other datasets. Can you suggest why?

3. L174: "The sample size in this dataset is larger than that of well-known databases of Glottolog and Ethnologue". This is misleading, Glottolog etc have >7000 languages. I think this means that the *dataset* has more varieties (n=100) than those named as Kra-Dai in Glottolog/Ethnologue. Please reword.

4. "we did not apply the Pseudo-Dollo model, because this model does not intrinsically agree with the situation of language evolution in our KD database". Please expand on what this means and why this model does not 'intrinsically agree'?

5. L1967: typo "TREEANNOATOR"

RESPONSE TO REVIEWERS' COMMENTS

Reviewer #1 (Remarks to the Author):

I have no further comments

Response: We sincerely appreciate your previous comments.

REVIEWER #2, comments:

I think the authors did an excellent job responding to my comments. I especially appreciate how they acknowledge theoretical and methodological issues that still need to be addressed, and how the results should be interpreted critically. I only have a couple of comments that I think the author should pay attention to.

Response: We sincerely appreciate your comments and acknowledgments of all our efforts.

- p.6 Status of Kra as a primary branch of Kra-Dai. The author explains that the observation that Kra is a primary branch is "consistent with the findings of previous historical linguistic works." The authors only on a small set of outdated and rather speculative/very preliminary studies. There is a lot more uncertainties and more diversity of opinion in the field of KD historical linguistics. The authors should check out Norquest's chapter and references therein. <https://www.degruyter.com/document/doi/10.1515/9783110558142-013/html?lang=en>

Response: According to Norquest's chapter, we summarized diverse opinions about the classification of Kra in section 2.1 of Supplementary Information, Lines 283-288, as follows:

'The first controversy is the position of Kra. Most linguists are in favor of Kra as a primary branch in the Kra-Dai language phylum such as Liang and Zhang (1996), Edmondson and Solnit (1997), Diller (2008), Chamberlin (2016), Ostapirat (2017) and Li (2019). However, Ostapirat (2005) proposed an original bifurcation between the Northern and Southern groups at an early stage, which demoted Kra to the subbranch of the Northern group. This classification was further revised by Norquest (2021), who demoted Kra to the position below the Kam-Sui group and as a sister of the Hlai-Tai branch.'

Notably, these classifications were relying on traditional linguistic comparative methods and would vary among these scholars based on the accumulated linguistic materials and different documents. In our study, the quantitative result of phylogenetic reconstruction showed that Kra should be a primary branch of Kra-Dai. This result was consistent with one situation of language classification proposed in previous studies (Liang and Zhang, 1996; Edmondson and Solnit, 1997; Diller *et al.*, 2008; Chamberlin, 2016; Ostapirat, 2017; Li, 2019).

References

1. Liang, M. & Zhang, J. R. *An Introduction to the Kam-Tai Languages (Chinese version)*. (China Social Sciences Press, 1996).
2. Edmondson, J. A. & Solnit, D. B. *Comparative Kadai: the Tai branch*. (Summer Institute of Linguistics Publications in Linguistics, 1997).

3. Diller, A., Edmondson, J. & Luo, Y. *The Tai-Kadai Languages*. (Routledge, 2008).
4. Chamberlain, J. R. Kra-Dai and the proto-history of South China and Vietnam. *The Journal of the Siam Society* **104**, 27-77 (2016).
5. Ostapirat, W. Kra-Dai in Southern China (Invited talk in the Nankai University). (Nankai University, 2017).
6. Li, P. J.-k. Dongdai yuzu de zujudi, kuosan ji qi shiqianwenhua (Chinese version). *Journal of East Linguistics* (2019).
7. Ostapirat, W. in *The Peopling of East Asia: Putting together archaeology, linguistics and genetics, Kra-dai and Austronesian: notes on phonological correspondences and vocabulary distribution* 107-131 (Routledge, 2005).
8. Norquest, P. in *The Languages and Linguistics of Mainland Southeast Asia* (eds Sidwell Paul & Jenny Mathias) 225-246 (De Gruyter Mouton, 2021).

- p.9 Assessing the results with existing linguistic knowledge. The author has done a good job. A very minor point is the use of "shared cognate sets" to mean set of forms inherited from a common ancestor or borrowing from other languages to be misleading. In historical linguistics "cognates" only means the former. To refer to both types of forms, something like "potential cognates" are preferable.

Response: We agree with your comment on the definition of cognate in linguistics. In the revision, we corrected the usage of “cognates” using “potential cognates” in the revision.

- p.13. The author assumes that Xing (1989) and Zhou and Luo (2001) are correct in classifying Yuanjiang as SWT. This is not the position linguists outside of Chinese take. Phonological innovations (as well as lexical innovations) in Yuanjiang show clearly that it is not part of SWT, despite it being "politically" classified as a Dai dialect. One of the many telling traits is the change /h-/ in reflex in the word for 'eye' (and other words with the same initial consonants in Proto-Tai) in contrast to /t-/ in SWT languages, cf. Li (1960)'s tentative classification of Tai languages among others.

Response: Thank you for your comments. To explicitly determine the low-level classification of Tai languages is an important work in linguistic studies, especially for language classification. In our work, we provisionally adopted the classification of Xing (1989) and Zhou and Luo (2001) that classifying Yuanjiang as Southwestern Tai. We respect your opinion on the classification of Yuanjiang. Therefore, we added your opinion and other differences between Yuanjiang and Southwestern Tai in the Supplementary Information, section 1.3 “*The classification and labeling of Kra-Dai languages*”, as follows:

“We labeled our Kra-Dai language samples followed by *Glottolog*, *Ethnologue*, and the references listed in Table S8. With the accumulation of language documents, the classifications of some Kra-Dai language samples required more detailed verification. For example, above Swadesh 100, the arguments of the classification of *Tai Ya* (Honghe in Xing 1989) and *Tai la* (Yuanjiang in Zhou and Luo 2001) were still matter in the view of historical linguistics. In our language samples, *TswYuanjiang* (also named *Tai Ya*) was classified as a Dai dialect and was part of the Southwestern Tai group. However, some linguistic materials inferred that Yuanjiang might share more phonological innovations with Central Tai rather than Southwestern Tai, and thus should be classified as Central Tai (Gedney and Thomas 1995; Zhang 1999; Hudak 2008; Pittayaporn 2009).

For example, the /k^h/ in the word ‘stream’ and ‘laugh’ of Yuanjiang was consistent with that of Central Tai. The /ts^h/ in the word ‘shower’, ‘six’ and ‘ear’ of Yuanjiang was more similar to the /tɕ^h/ of Central Tai rather than the /h/ of Southwestern Tai. The change of /h-/ in the word ‘eye’ of Yuanjiang was distinct from /t-/ in Southwestern Tai (Li 1960). Despite the evidence, in this study, we provisionally adopted the classification of Xing (1989) and Zhou and Luo (2001) that Yuanjiang was classified in Southwestern Tai. Such classification was also adopted by the widely-used linguistic database of Glottolog (Glottocode: taih1246) and Ethnologue (ISO 639-3 code: tiz).

In summary, several fundamental questions on Kra-Dai language classifications deserved further examination based on more traditional linguistic comparative studies. Accordingly, it allowed us to obtain more linguistic materials from diverse linguistic perspectives such as phonology and grammar.”

We also highlighted that several fundamental questions on Kra-Dai language classifications deserved further examination based on traditional linguistic comparative study. Accordingly, it allowed us to obtain more linguistic materials from diverse perspectives such as phonology and grammar.

References:

1. Xing, G. W. *Upper Hongjin Dai Ya Language (Chinese version)*. Language Publishing House (1989).
2. Zhou, Y. W. & Luo, M. Z. *A Study of Dai Dialects (Chinese version)*. Ethnic Publishing House (2001).
3. Gedney, W. J., and Thomas J. H. *William J. Gedney's central Tai dialects: glossaries, texts, and translations*. Michigan papers on South and Southeast Asia, no. 43. Ann Arbor, Mich: Center for South and Southeast Asian Studies, University of Michigan (1995).
4. Zhang, J. R. *Zhuangyu Fangyan Yanjiu (Chinese version)*. (Sichuan Minzu Publishing House, 1999).
5. Hudak, T. J. (Ed.). *William J. Gedney's Comparative Tai source book*. Honolulu: University of Hawaii Press (2008).
6. Pittayaporn, P. *The Phonology of Proto-Tai*. Department of Linguistics, Cornell University (2009).
7. Li, F. K. A tentative classification of Tai dialects. *Culture in history: Essays in honor of Paul Radin*, 951-959 (1960).

Reviewer #3 (Remarks to the Author):

This is a very promising paper that provides an integrated perspective into the evolution and dispersal of the Kra-Dai language family. I enjoyed reading this and think it will become an important paper.

Response: We sincerely appreciate your recognition of our work.

The authors are to be commended for making their data, analysis scripts, code and XML all available.

Response: Open-source codes and data are what we've been working on.

I have two major recommendations one regarding the use of the inaccurate HME estimator for

model comparison, and one regarding the lack of detail about the integration genetic and archaeological analysis. The remainder of my recommendations are not as critical but would be nice to have.

I recommend this paper be accepted after minor revisions.

Major Recommendations:

1. Provide details on the model comparison, and fix if the HME in Tracer was used.

I could not find any details on how the phylogenetic model comparison was done, beyond the authors calculating Bayes Factors. How was the marginal likelihood estimated? Looking at the supplementary table, it suggests that the authors used the Harmonic Mean Estimator implemented in Tracer. However the HME is known to provide inaccurate results as it is severely biased and substantially overestimated the true marginal likelihood (see Xie et al. 2011, Sys. Biol) which may mean that the model chosen as best is not the best. Current best practice is to use a formal model comparison method like Path Sampling (e.g. Beale et al. 2013, Mol. Biol. Evol) or Nested Sampling (e.g. Russel et al. 2018, Sys. Biol).

Response: Tracer v1.6 provided the Harmonic Mean Estimator in the *Model Comparison* option of the *Analysis* window, and this was exactly how we compared the models. According to your comments, we used an alternative approach of Path Sampling to compare various models with different parametric combinations. Path sampling was performed with the model-selection package (v1.5.3) implemented in the BEAST software. The XML files used to perform path sampling were generated following the guideline in <http://www.beast2.org/2014/07/14/path-sampling-with-a-gui>. Specifically, we set the alpha to 0.3, the number of steps to 8, the chain length to 500,000, the pre-burn to 500,000, and the burn-in to 50% to ensure the sum of Effective Sample Size (sum(ESS)) > 200. The comparison results of the Path Sampling approach were listed below and were added as a table into Supplementary Table S3, sheet name 'Path Sampling'. The results showed that the best model combination was the Covarion model with the relaxed Lognormal clock model, which was consistent with the observations using the HME approach. In the revision, we added a brief description of this approach in Lines 373-374 in the section of Methods: "*The comparison of model combinations was also performed by Path Sampling method following the guideline (URL: <http://www.beast2.org/2014/07/14/path-sampling-with-a-gui>) (Table S3).*" Additionally, the above details for using Harmonic Mean Estimator and Path Sampling approaches were added in the two sheets of Supplementary Table S3, respectively.

Table R1. Statistical comparisons of models with different parametric combinations using the Path Sampling approach. The Marginal likelihood values estimated by the path sampling method (Baele, *et al.* 2013) were used to compare. The higher value indicated a better model fit. Differences between marginal likelihoods (specifically, Bayes factors) are reported. Positive values indicate a better relative model fit of the row's model compared to the column's model.

No.	Model	Marginal L estimate	sum(ESS)	1	2	3	4	5	6
1	Covarion_Relaxed_LogNormal_Clock	-9117.84	343.7178	-	144.49	355.54	106.1	212.45	188.05
2	Covarion_Strict_Clock	-9262.33	407.1323	-144.49	-	211.05	-38.39	67.96	43.56
3	CTMC_Relaxed_LogNormal_Clock_1Gamma	-9473.38	247.4268	-355.54	-211.05	-	-249.44	-143.09	-167.49
4	CTMC_Relaxed_LogNormal_Clock_4Gamma	-9223.94	253.0932	-106.1	38.39	249.44	-	106.35	81.95
5	CTMC_Strict_Clock_1Gamma	-9330.29	370.5614	-212.45	-67.96	143.09	-106.35	-	-24.4
6	CTMC_Strict_Clock_4Gamma	-9305.89	450.9309	-188.05	-43.56	167.49	-81.95	24.4	-

Reference:

1. Baele, G. *et al.* Improving the Accuracy of Demographic and Molecular Clock Model Comparison While Accommodating Phylogenetic Uncertainty. *Molecular Biology and Evolution* **29**, 2157-2167 (2012).
2. Integrate information about the mitochondrial and archaeological analyses into main text:

The mtDNA and archaeological analysis is interesting and really bolster the author's arguments about the underlying demographic dispersal. However, these are not mentioned in the main text and it was only when I read the methods that I realised these were there. Please integrate these findings into the main narrative as without them the final parts of the discussion read as far more speculative than they are (and I was going to flag this as one of my major complaints until I found these details in the methods/results).

Response: Thank you for your comments. We supplemented more descriptions for results of mtDNA and archaeological analysis in the sections of *Results* and *Discussion* in the revision. The followings were added in section *Results*, Lines 201-228:

“To provide a more comprehensive understanding of the social and cultural context surrounding the Kra-Dai language divergence and dispersal, we integrated interdisciplinary evidence from genetics, archaeology, paleoecology, and paleoclimatology to depict the evolutionary process of Kra-Dai languages. As illustrated in Figure 3, the divergence tempo of Kra-Dai languages showed that the initial divergence occurred at ~4,000 years BP and the second one occurred at ~3,200 years BP, then the language numbers increased continuously in the past 2,300 years (Figure 3A, Figure S10). According to archaeological evidence, the number of archaeological sites in Southern China decreased dramatically at ~4,000 years BP, then increased and reached its maximum at ~3,000 years BP (Figure 3B). The genetic evidence was represented by the Bayesian Skyline Plot of the Kra-Dai mtDNA lineages which reflected the historical change of Kra-Dai population size. Generally, we found two phases of population growth, of which the former was an approximately 17-fold demographic increase during 6,400 – 4,200 years BP and the latter was an approximately 16-fold demographic leap from 3,500 years BP till now (Figure 3C). In addition, the paleo-ecological evidence suggested that the survival probabilities of tropical rice decreased dramatically in eastern China and high-altitude southwestern China during 4,400 – 3,500 years BP and then maintained a relatively stable³⁹ (Figure 3D). Lastly, based on the paleo-climatological evidence⁴⁰⁻⁴², we found the global temperature decrease known as the 4.2K event, which took place from 4,400 to 3,500 years BP and minimum at ~4,000 years BP. Then, the global temperature became relatively stable in the past 3,000 years (Figure 3E).

Accordingly, we could summarize the evolutionary history of Kra-Dai languages into three periods. The first one was the “contraction period” during the 4.2K event (4,400 – 3,500 years BP), coupled with the initial divergence of Kra-Dai languages, a nearly unchanged population size, decreasing archaeological sites, survival probabilities of rice, and temperature. The second one was

the “recovery period” after the 4.2K event (3,500 – 2,300 years BP), corresponding to the early divergence events of Kra-Dai languages, a more temperate climate than before, and a steady increase of archaeological sites and population size. The third one was the “prosperity period” (2,300 years BP - now), which witnessed a rapid increase in language numbers and population size (Supplementary Information section 2.8).”

The followings were added in section *Discussion*, Lines 279-304:

“Furthermore, we observed the strong coupling of the linguistic and demographic dynamics with the changes in the paleoenvironmental context (Figure 3 and Figure S10). In general, the paleoenvironmental context consists of the paleoecologic and paleoclimatic factors which are regarded as crucial drivers to shape the demographic activities of prehistoric populations⁵⁰⁻⁵². Synthesizing the interdisciplinary evidence, we proposed a possible scenario that prehistoric Kra-Dai language divergence and dispersal accompanying population expansion could be driven by the dynamic changes in the paleoenvironmental context (Supplementary Information section 2.7 and section 2.8). In particular, as early as 5,000 years BP, the rice farmers in the lower Yangtze River Valley, namely, Bai Yue nationalities, were divided into Kra-Dai-speaking and Austronesian-speaking populations, respectively^{1,2,26,44-46,53,54}. During the “contraction period”, the Kra-Dai-speaking populations were forced to experience the migration process and population divergence in the deteriorating environment. This process induced the initial Kra-Dai language divergence. Due to the collapse of agriculture and the shortage of food, some settlements were abandoned, resulting in a decrease in archaeological sites; meanwhile, the number of Kra-Dai-speaking populations of maternal lineages grew slowly, indicating that the population size might maintain nearly unchanged. In the “recovery period”, the temperature did not fluctuate dramatically, and food production became more stable than before. This situation promoted the steady growth of the population size of Kra-Dai-speaking populations, and people started to migrate actively and more frequently to find more settlements. Such population activities in the “recovery period” also resulted in the early language divergence events. These findings suggested that the prehistoric divergence of Kra-Dai languages might be coupled with the climate-induced demographic activities (e.g., migration) of Kra-Dai-speaking populations. In contrast, during the “prosperity period”, the long-term stable temperature and food production allowed the number of Kra-Dai languages and the size of Kra-Dai-speaking populations to increase spontaneously, contributing to more frequent demographic activities such as population expansions and interactions. (Figure 3, Figure S10, Supplementary Information section 2.7, and section 2.8).”

3. As it stands, the current manuscript spends a LOT of time talking about one minor language, Saek. I appreciate the care given to evaluation of the Saek language but takes up a lot of space in main text. If space is an issue then I think that much of this Saek related content could be moved into the supplement (with an appropriate callout in main text), to enable the interdisciplinary results narrative to be highlighted (much more interesting than one language).

Response: Thank you for your comments. We used the Saek language as an example not only to evaluate its authentic classification but also to critically discuss Bayesian phylo-linguistics and the results. Therefore, we suggested that this part could be preserved in the main text. To highlight the interdisciplinary results, we added more related content in the revised version of the main text as our shown in response 2.

Minor recommendations:

4. Please provide a fuller discussion of the preferred homeland. The best-fitting homeland location only captures 47% of the probability mass, while two other homelands (Guizhou and Yunnan) have appreciable probabilities of ~20%. Given that these are also potentially likely candidates for homelands for this family, please provide some discussion of this and how -- if at all -- changing this homeland would affect the downstream interpretations in the paper.

Response: Using the phylogeographic approach, we reconstructed the ancestral character state at the root of given the Kra-Dai language trees and estimated the probabilities of five distinct areas with considerations of phylogenetic uncertainties. For NULL hypothesis of the phylogeographic model, we could observe 20% for each distinct area which was inferred as a dispersal center equiprobably. Here, our results showed that the Guangdong-Guangxi coastal area should be a dispersal area with a maximum probability of 47.0%, which was much higher than 20%. In addition, the MCMC method would give us a set of probabilities for each area, which could be tested by the Wilcoxon signed rank test to find whether there are significant differences among the probabilities of the five distinct areas in the posterior samples. We also used a bar plot to show the values and standard deviation of probabilities (Figure R1). Accordingly, we suggested that the coastal area should be the dispersal center of Kra-Dai languages with a significantly higher probability than other areas. We added the corresponding contents in the revised Supplementary Information section 2.4, Lines 421-428, as follows:

“For the NULL hypothesis of the phylogeographic model, we could observe 20% for each distinct area which was inferred as a dispersal center equiprobably. Here, our phylogeographic reconstructions indicated that the most likely dispersal center of Kra-Dai languages was in the coastal areas of China with a maximum probability of 47.0%, which was much higher than 20%, as shown in Table S4. In addition, we used the Wilcoxon signed rank test to find that the probability of the coastal area was significantly higher than those of the other four distinct areas. Accordingly, we suggested that the coastal area should be the homeland of Kra-Dai languages with a significantly higher probability than other areas (Figure S7).”

Figure R1. Probabilities of geographical distribution for the root of Kra-Dai languages. Bars represented the standard deviation. The significance tested by Wilcoxon signed rank test was indicated by symbol “*”, “***” represented $p < 0.001$.

5. The authors say they use the linguistic comparative method and there are some nice examples in the supplementary material of where they have done this. However, the state of Kra-Dai historical linguistics is not as well-established as many other families. Can the authors provide a table of their established sound correspondences to enable future work to extend and build off their work here?

Response: The reconstructions of proto-language and regular sound correspondences are essential in traditional historical linguistics. In practice, we referred to the reconstructed proto-forms of etyma (or cognates) for the Proto-Kra-Dai language and its five branches from different bibliographies. These bibliographies not only provided the proto-forms but also enabled us to summarize sound correspondences. The referred bibliographies are listed as follows:

Proto-Kra-Dai: Liang and Zhang (1996), Peiros (1998);

Proto-Kra: Ostapirat (1999);

Proto-Kam-Sui (including Proto-Kam and Proto-Sui): Thurgood (1988), Edmondson and Yang (1998), Zeng (1994), Ferlus (1996), Peiros (1998), Huang (2002), Ostapirat, (2006), Long (2018);

Proto-Lakkia: L-Thongkum (1992);

Proto-Ong-Be: Chen (2018);

Proto-Hlai: Matisoff (1988), Ostapirat, (2004), Norquest (2007);

Proto-Tai: Li (1977), Ferlus (1990), Pittayaporn (2008 and 2009), Ostapirat (2013)).

Establishing the full table of sound correspondence is a critical and large project for traditional linguistic comparison which allowed us to compile a greater size of basic vocabulary set than those of Swadesh wordlists. This project is currently underway by one of our authors, Jiaqi Ge. As a part of his doctoral thesis, he is making great efforts to complete the full table of sound correspondences for Kra-Dai languages using a great vocabulary list rather than the wordlist in this study and

following the standard traditional historical comparison. And the full table will be finished and available with the publication of his doctoral thesis and another journal paper related to the historical linguistic comparison of Kra-Dai languages.

References:

1. Liang, M. & Zhang, J. R. *An Introduction to the Kam-Tai Languages (Chinese version)*. (China Social Sciences Press, 1996).
2. Peiros, I. *Comparative Linguistics in Southeast Asia*. (Pacific Linguistics, Research School of Pacific and Asian Studies, 1998).
3. Ostapirat, W. *Proto-Kra*. (University of California, Berkeley, 1999).
4. Thurgood, G. Notes on the reconstruction of Proto-Kam-Sui. *Comparative Kadai: linguistic studies beyond Tai*, 179-218 (1988).
5. Edmondson, J. A. & Yang, Q. Word - initial Preconsonants and the History of Kam-Sui Resonant Initials and Tones. *Comparative Kadai: Linguistic studies beyond Tai*, 143-166 (1988).
6. Zeng, X. Y. *Hanyu Shuiyu Guanxici Yanjiu (Chinese version)*. (Chongqing Publishing Group, 1996).
7. Ferlus, M. Remarques sur le consonantisme du proto kam-sui. *Cahiers de linguistique-Asie Orientale* **25**, 235-278 (1996).
8. Huang, Y. *Hanyu Dongyu Guanxici Yanjiu (Chinese version)*. (Tianjin Guji Press, 2002).
9. Ostapirat, W. Alternation of tonal series and the reconstruction of Proto-Kam-Sui. *Dah-an Ho, H. Samuel Cheung, Wuyun Pan, & Fuxiang Wu(eds.), Linguistic studies in Chinese and Neighboring languages: Festschrift in Honor of Professor Pang-Hsin Ting on His 70th Birthday*, 1077-1121 (2006).
10. Long, R.-T. Research on the Tone Merger of Kam Language (Chinese version). *Guizhou Ethnic Studies* **39**, 194-199 (2018).
11. L-Thongkum, T. A Preliminary reconstruction of Proto-Lakkja (Cha Shan Yao). *The Mon-Khmer Studies Journal* **20**, 57-90 (1992).
12. Chen, Y.-l. *Proto-Ong-Be*, University of Hawai'i at Manoa, (2018).
13. Matisoff, J. A. Proto-Hlai initials and tones: a first approximation. *Comparative Kadai: linguistic studies beyond Tai*, 289-321 (1988).
14. Ostapirat, W. Proto-Hlai sound system and lexicons. *Studies on Sino-Tibetan languages: Papers in honor of Professor Hwang-cherng Gong on his seventieth birthday*, 121-175 (2004).
15. Norquest, P. K. *A phonological reconstruction of Proto-Hlai*. (The University of Arizona, 2007).
16. Li, F. K. *A handbook of comparative Tai*. (The University Press of Hawaii, 1977).
17. Ferlus, M. REMARQUES SUR LE CONSONANTISME DU PROTO THAI-YAY (Révision du proto tai de LI Fangkuei). *23rd International Conference on Sino-Tibetan Languages and Linguistics* (1990).
18. Pittayaporn, P. Proto-Southwestern-Tai revised: A new reconstruction. *JSEALS*, 119 (2009).
19. Pittayaporn, P. *The phonology of proto-Tai*, Cornell University, (2009).
20. Ostapirat, W. The rime system of Proto-Tai. *Bulletin of Chinese Linguistics* **7**, 189-227 (2013).

Comments and Suggestions:

6. The authors abbreviation "KD" makes the paper more opaque, all to save 4 letters. It's unnecessary,

please remove.

Response: In the revision, we replaced the abbreviation "KD" using "Kra-Dai".

7. The link to 4.2k event is interesting, but the Effective Population size from the mitochondrial genomes appears to show no effect of the 4.2k event, unlike the other datasets. Can you suggest why?

Response: Before the 4.2K event, the environment is suitable for living, so we could observe an approximately 17-fold demographic increase during 6,400 – 4,200 years BP. During the 4.2K event, the deteriorating environment led to a shortage of food and population migration. In addition, the changes in the environment would not result in sudden population reductions that were usually caused by large-scale wars or diseases because the interactions among populations were limited in the late Neolithic period. Under such circumstances, it seems plausible that the population size could hardly increase rapidly. Then, after the 4.2K event, the environment became stable and thus allowed the population size to grow (an approximately 16-fold demographic leap from 3,500 years BP till now). In summary, the 4.2K event reduced the growth rate and kept the population size nearly unchanged.

8. L174: "The sample size in this dataset is larger than that of well-known databases of Glottolog and Ethnologue". This is misleading, Glottolog etc have >7000 languages. I think this means that the *dataset* has more varieties (n=100) than those named as Kra-Dai in Glottolog/Ethnologue. Please reword.

Response: Thank you for your comments. We modified the corresponding sentences in Lines 154-155, "*The sample size in this dataset is larger than that of the languages named as Kra-Dai or Tai-Kadai in Glottolog and Ethnologue databases.*"

9. "we did not apply the Pseudo-Dollo model, because this model does not intrinsically agree with the situation of language evolution in our KD database". Please expand on what this means and why this model does not 'intrinsically agree'?

Response: The Pseudo-Dollo model assumes an evolutionary process that once a cognate is gained, it could only be lost once and never be gained again (Bouckaert and Robbeets, 2017). This model is neither time-reversible nor in an equilibrium state as the evolution is directional (Hoffmann et al., 2021). And this model can also be formulated as a three-state continuous time Markov chain (CTMC) model (Bouckaert and Robbeets, 2017). The model's assumption might not apply to the complicated linguistic *sprachbund* in MSEA where frequent language contact might occur. As we indicated in Section 2.1 of Supplementary Information, '*we excluded the loanwords borrowed from Middle Chinese and Modern Chinese dialects, but maintained the obscure loanwords borrowed from Old Chinese and other non-Sinitic languages*', the lexical borrowings among Kra-Dai languages might also violate the assumptions of Pseudo-Dollo model (e.g., the Saek language).

Accordingly, we supplemented the reason for not applying the Pseudo-Dollo model in Lines 200-203 in Supplementary Information section 1.4:

"We did not apply the Pseudo-Dollo model because this model assumes that once a cognate is gained, it could only be lost once and never be gained again, which did not intrinsically agree with the complicated and contact-frequent language evolution scenario in our Kra-Dai database."

References:

1. Bouckaert, R. & Robbeets, M. Pseudo Dollo models for the evolution of binary characters along a tree. *BioRxiv*, 207571, doi:10.1101/207571 (2017).
2. Konstantin Hoffmann and others, Bayesian phylogenetic analysis of linguistic data using BEAST, *Journal of Language Evolution*, **6** (2): 119–135, <https://doi.org/10.1093/jole/lzab005> (2021)

10. L1967: typo "TREEANNOATOR"

Response: Thank you for pointing it out. We corrected this typo in the revision.

REVIEWERS' COMMENTS

Reviewer #3 (Remarks to the Author):

I thank the authors for their careful responses to my comments, and now recommend this paper be accepted.

RESPONSE TO REVIEWERS' COMMENTS

Reviewer #3 (Remarks to the Author):

I thank the authors for their careful responses to my comments, and now recommend this paper be accepted.

Response: We sincerely appreciate your previous comments.